# MALT-1 mediates IL-17 neural signaling to regulate *C. elegans* behavior, immunity and longevity

Sean M. Flynn ● [1], Changchun Chen ● [1,3], Murat Artan[1], Stephen Barratt[1], Alastair Crisp[1], Geoffrey M. Nelson ● [1,4], Sew-Yeu Peak-Chew[2], Farida Begum[2], Mark Skehel[2] & Mario de Bono ● [1,5✉]

Besides pro-inflammatory roles, the ancient cytokine interleukin-17 (IL-17) modulates neural circuit function. We investigate IL-17 signaling in neurons, and the extent it can alter organismal phenotypes. We combine immunoprecipitation and mass spectrometry to biochemically characterize endogenous signaling complexes that function downstream of IL-17 receptors in *C. elegans* neurons. We identify the paracaspase MALT-1 as a critical output of the pathway. MALT1 mediates signaling from many immune receptors in mammals, but was not previously implicated in IL-17 signaling or nervous system function. *C. elegans* MALT-1 forms a complex with homologs of Act1 and IRAK and appears to function both as a scaffold and a protease. MALT-1 is expressed broadly in the *C. elegans* nervous system, and neuronal IL-17–MALT-1 signaling regulates multiple phenotypes, including escape behavior, associative learning, immunity and longevity. Our data suggest MALT1 has an ancient role modulating neural circuit function downstream of IL-17 to remodel physiology and behavior.

---

[1] Cell Biology Division, Medical Research Council Laboratory of Molecular Biology, Cambridge CB2 0QH, United Kingdom. [2] Biological Mass Spectrometry and Proteomics, Cell Biology Division, Medical Research Council Laboratory of Molecular Biology, Cambridge CB2 0QH, United Kingdom. [3] Present address: Umeå Center for Molecular Medicine, Wallenberg Center for Molecular Medicine, Umeå University, SE-901 87 Umeå, Sweden. [4] Present address: Department of Biomedical Informatics, Harvard Medical School, Boston, MA 02115, USA. [5] Present address: Institute of Science and Technology Austria (IST Austria), Am Campus 1, 3400 Klosterneuburg, Austria. ✉email: mdebono@ist.ac.at

mmune signaling pathways can regulate the development and function of the nervous system in both health and disease[1–3]. Many of these effects are mediated by cytokines, small, secreted proteins that can participate in neuroimmune and inter-neuronal communication. For example, low levels of IL-1β and TNFα regulate synaptic and homeostatic plasticity in healthy animals[4,5]; pathological levels of proinflammatory cytokines during inflammation can disrupt fetal brain development, alter adult behavior[6–9], and drive hyperalgesia and neuroinflammatory diseases[10]. Progression of neurodegenerative diseases, including Alzheimer's, Parkinson's and Amyotrophic lateral sclerosis (ALS), has also been associated with chronic inflammation[11,12].

Recent work shows that the interleukin 17 (IL-17) pro-inflammatory cytokine can modify neural circuit activity. In a rodent model of infection during pregnancy, IL-17 secretion during maternal immune activation drives autism-related behaviors in the pups[13]. This phenotype is associated with hyper-activity of a specific cortical sub-region that expresses IL-17 receptors (IL-17R)[14]. In mice, IL-17 can also lower the activation threshold of nociceptive neurons, and contributes to mechanical hyperalgesia[15,16]. In C. elegans IL-17Rs are expressed throughout the nervous system, and ILC-17.1 (interleukin cytokine 17 related 1), a homolog of mammalian IL-17s, has been shown to act on the RMG hub interneurons, increasing their response to pre-synaptic input from oxygen ($O_2$) sensors. The increased circuit gain conferred by ILC-17.1 enables C. elegans to persistently escapes 21% $O_2$, an aversive cue associated with surface expo-sure[17]. Specific sensory responses and behaviors are thus modu-lated by IL-17 across distantly-related species, suggesting IL-17 has broad and conserved roles in regulating neuronal properties.

While IL-17's action on the nervous system is now established, its molecular effectors there are poorly understood. Moreover, the extent to which IL-17 signaling contributes to brain function and physiology is unclear, even in the well-defined C. elegans nervous system.

Here, we report that IL-17 signaling in the C. elegans nervous system is mediated by the paracaspase MALT-1. MALT1 is an ancient protein[18] studied extensively, and almost exclusively, in the mammalian immune system. It is a key signaling molecule in innate and adaptive immunity, mediating signaling from ITAM-containing (immunoreceptor tyrosine-based activation domain) receptors, including the B-cell and T-cell receptors[19–21]. MALT1 has not been shown to mediate IL-17 signaling, but there has been speculation of such involvement. In situ hybridization suggests widespread MALT1 expression in mouse brain, (Allen Brain Atlas), but no physiological role in neurons has been reported. We find that C. elegans MALT-1 is expressed throughout the nervous system and forms an in vivo complex with IL-17 signaling components, namely the C. elegans homo-logs of Act1, IRAK and IκBζ/IκBNS. We show that MALT-1 acts both as a protease and a scaffold to regulate neural function. Defects in IL-17/MALT-1 signaling lead to reconfigured gene expression, and changes in behavior and physiology, including altered immunity and extended lifespan.

## Results

### Proteomics identifies an ACTL-1–IRAK–MALT-1–NFKI-1 complex.
C. elegans IL-17 signaling components appear to be expressed predominantly in the nervous system[17]. We epitope tagged all soluble IL-17 pathway components highlighted by genetics[17], immunoprecipitated them from C. elegans extracts, and identified interacting proteins using mass spectrometry (MS, Fig. 1a).

ACTL-1 and PIK-1 are C. elegans orthologs of mammalian Act1 and IRAKs, respectively, and signal downstream of the

C. elegans IL-17 co-receptors ILCR-1 and ILCR-2[17]. Genetic analysis suggests NFKI-1, a homolog of mammalian IκBζ and IκBNS, acts downstream of ACTL-1, PIK-1, and ILCR-1/ILCR-2 co-receptors[17].

We tagged endogenous ACTL-1 with a FLAG epitope, endogenous PIK-1 with a Myc epitope, and integrated an nfki-1::gfp transgene. We showed the tagged proteins were functional (Supplementary Fig. 1), and immunoprecipitated them from C. elegans extracts. As controls, we immunoprecipitated proteins unrelated to IL-17 signaling tagged with the same epitopes. Using mass spectrometry (LC-MS/MS) we identified specific interactors for each signaling component (Fig. 1b–g and Supplementary Data 1a–c).

As expected from co-IP experiments using mammalian tissue culture cells[17], PIK-1 co-precipitated specifically with ACTL-1 (Fig. 1b), and reciprocally, ACTL-1 co-precipitated specifically with PIK-1 (Fig. 1c). IP of NFKI-1 also identified ACTL-1 and PIK-1/IRAK as specific interactors, suggesting these proteins form a complex in vivo (Fig. 1d). We identified other apparently specific interactors for each component. These are listed in Supplementary Data 1 as a resource.

The C. elegans ortholog of the paracaspase MALT1 consistently co-immunoprecipitated with each of ACTL-1, PIK-1 and NFKI-1 (Fig. 1b–g). MALT1 paracaspases are cysteine proteases with specificity for arginine residues[23,24]. Their caspase-like protease domain is highly conserved, as is their domain organization, which consists of an N-terminal death domain (DD) followed by 2–3 Ig (immunoglobulin)-like motifs that flank the paracaspase domain (Supplementary Fig. 2a)[25]. Mammalian MALT1 signals downstream of B cell, T cell, and other cell surface receptors containing an ITAM motif, and forms a filamentous complex called the CBM signalosome that contains a CARD domain protein, BCL10, and MALT1[19–21] (Supplementary Fig. 2b). The functions of MALT1 in the immune system are under intense scrutiny, but its roles elsewhere, and in invertebrates, have not been established.

To confirm the biochemical interactions of MALT-1 with C. elegans IL-17 signaling components, we expressed functional, GFP-tagged MALT-1 pan-neuronally, and identified interacting partners using IP/MS of extracts from the transgenic C. elegans strain. As a control, we performed IP/MS on extracts from strains expressing GFP-tagged neuronal proteins unrelated to IL-17 signaling. ACTL-1, PIK-1, and NFKI-1 each interacted specifically with MALT-1-GFP (Fig. 1h, i). We also identified other specific MALT-1 interactors (Supplementary Data 1d) including the C. elegans ortholog of mammalian SARM1, called TIR-1, which is implicated in the immune response[26,27], left/right asymmetry of an olfactory neuron[28], and experience-dependent plasticity[29]. MALT-1 also interacted specifically with a large group of proteins implicated in RNA metabolism, including splicing factors and poly A binding proteins, suggesting it may localize to the nucleus or ribonucleoprotein particles (RNPs) (Supplementary Fig. 3).

**MALT-1 promotes aggregation and escape from 21% $O_2$.**
MALT1 has not previously been implicated in IL-17 signaling or neural function. In C. elegans, ILC-17.1 signals through the ILCR-1/ILCR-2 receptors on the RMG interneurons to increase RMG responsiveness to input from their pre-synaptic partner, the URX $O_2$-sensing neurons (Fig. 1j). Increased RMG signaling enables C. elegans to strongly and persistently escape 21% $O_2$ and to aggregate[17,30,31]. To probe the functional relevance of our pro-teomics data we sought malt-1 alleles in a collection of 583 strains isolated in a genetic screen for aggregation-defective mutants. This collection has been subjected to whole genome sequencing, and previously yielded IL-17 pathway mutants[17]. Four strains in

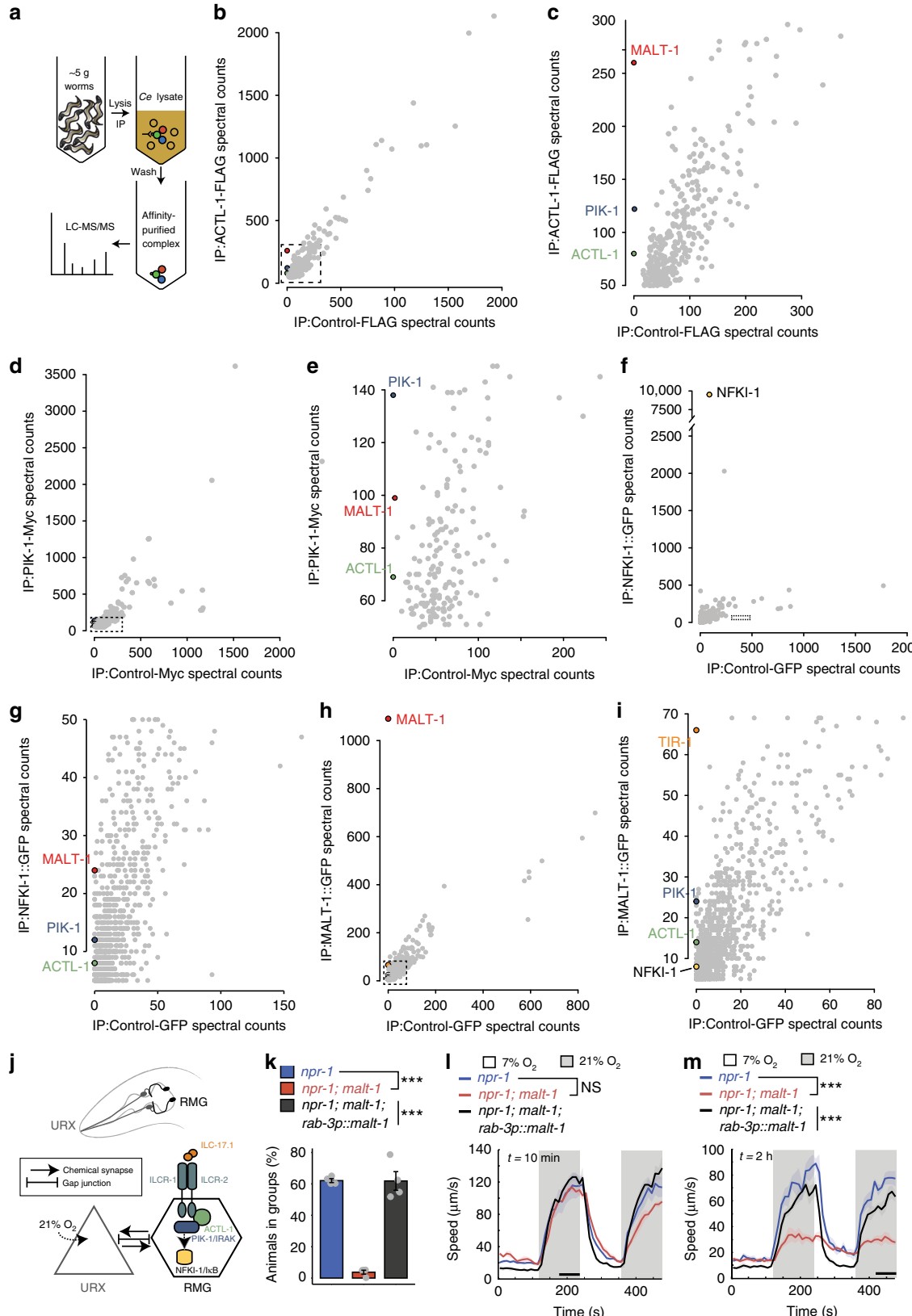

the collection harbored *malt-1* alleles; one introduced a premature stop codon; another mutated the highly conserved E464 residue (Supplementary Fig. 4a and b), which is essential for catalytic activity in mammalian MALT1[32]. We mapped the aggregation defect of this strain to an interval containing *malt-1* (Supplementary Fig. 4c). Targeted disruption of *malt-1* using

CRISPR/Cas9 resulted in an aggregation-defective strain whose phenotype could be rescued using a wild-type *malt-1* transgene (Fig. 1k; and Supplementary Fig. 4f). These data confirm that MALT-1, like IL-17 signaling, promotes aggregation.

*C. elegans* aggregate to escape 21% $O_2$, a signal of surface exposure[33–35]. In wild *C. elegans* isolates, 21% $O_2$ evokes

**Fig. 1 MALT-1 forms a complex with ACTL-1, PIK-1/IRAK, and NFKI-1. a** Schematic for affinity-purification and LC-MS/MS analysis of epitope-tagged IL-17 signaling components from *C. elegans* extracts. Ce = *C. elegans*. **b–i** Pull-down of ACTL-1-FLAG, PIK-1-Myc, or NFKI-1::GFP specifically co-IPs MALT-1 (**b–g**). Conversely, pull-down of MALT-1::GFP specifically co-IPs ACTL-1, PIK-1, and NFKI-1 (**h** and **i**). Total spectral counts, a semi-quantitative readout of abundance[22], are shown. **c**, **e**, **g**, and **i** as in **b**, **d**, **f**, and **h** except showing only the region marked by the black box in **b**, **d**, **f**, and **h**, respectively. **f–i** Data is representative of two (**f** and **g**), or three (**h** and **i**) biological replicates. **j** Schematic of IL-17 signaling in the $O_2$-escape circuit. Increases in $O_2$ levels are sensed by URX neurons, which tonically signal to RMG hub interneurons. IL-17 signaling increases the responsiveness of RMG neurons to promote escape from 21% $O_2$. **k** *malt-1* promotes *C. elegans* aggregation (N = 4 assays). Data are presented as mean values +/− SEM. ***P < 0.001, one-way ANOVA with Tukey's post hoc HSD. **l** and **m** *malt-1* mutants are strongly aroused by 21% $O_2$ if stimulated immediately after transfer to the assay plate (**l**), but respond weakly to 21% $O_2$ if allowed to settle over a 2 h period (**m**). **l** n = 86 animals (*npr-1*), n = 46 animals (*npr-1; malt-1*), n = 53 animals (*npr-1; malt-1; rab-3p:: malt-1*). **m** n = 46 animals (*npr-1*), n = 72 animals (*npr-1; malt-1*), n = 46 animals (*npr-1; malt-1; rab-3p::malt-1*). Plots show average speed (line) and SEM (shaded regions). Time of assay after transfer is shown at top left. NS, P = 0.8, ***P < 0.001, two-sided Mann-Whitney U test. Here and in subsequent figures, black bars indicate time intervals used for statistical comparisons. See also Supplementary Figs. 1–4 and Supplementary Data 1.

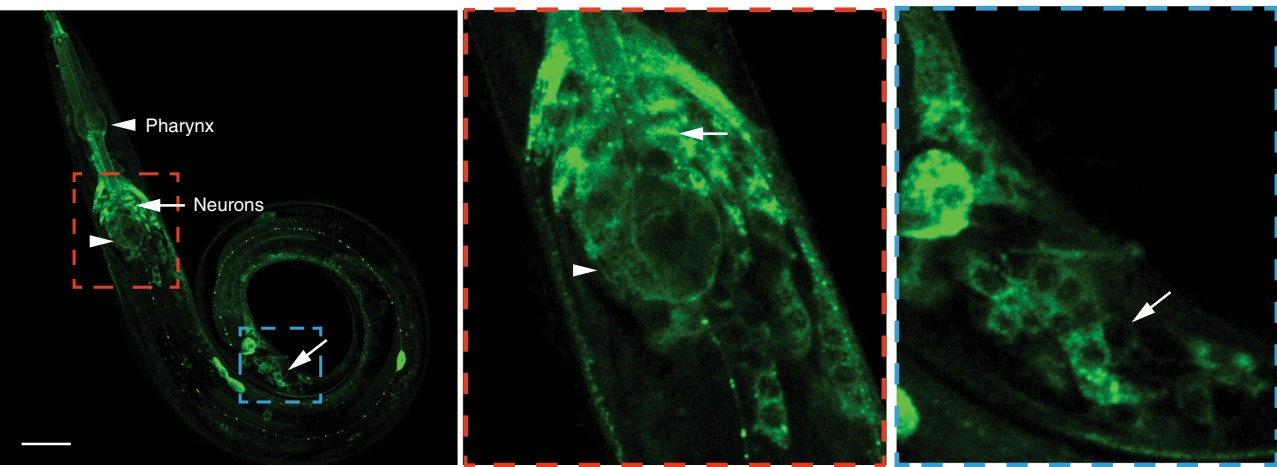

**Fig. 2 MALT-1 is expressed widely in the nervous system.** A transgene expressing C-terminally GFP-tagged MALT-1 from its endogenous promoter (4 kb of upstream DNA) is expressed broadly in the nervous system, including many neurons in the head (red box) and tail (blue box). MALT-1::GFP expression is also seen throughout the pharynx. Similar results were obtained in 3 experiments. White arrows point to neurons, arrowheads point to the pharyngeal bulbs. Scale bar: 20 µm. See also Supplementary Fig. 5.

sustained arousal[36], a response also observed in *npr-1 (neuropeptide receptor 1)* mutants of the domesticated N2 lab strain[37]. By contrast, *npr-1* mutants defective in IL-17 signaling are not aroused by 21% $O_2$ in the absence of an additional arousal stimulus (e.g., being picked), and if this requirement is met, the arousal evoked by 21% $O_2$ is not sustained[17]. *malt-1* mutants showed these hallmark phenotypes (Fig. 1l, m), consistent with MALT-1 playing a role in *C. elegans* IL-17 signaling.

*malt-1* mutants exhibited grossly normal growth rates, fertility, mating and feeding behaviors, and locomotion, although this was not quantitated. They exhibited a small but significant reduction in thrashing rate, suggesting a weak defect in locomotion (Supplementary Fig. 4e). Compared to their defects in escape from 21% $O_2$ however, this phenotype was relatively subtle.

**MALT-1 modulates responsiveness of RMG interneurons to $O_2$.** *malt-1::GFP* and *malt-1::RFP* transgenes were expressed broadly in the nervous system (Fig. 2), including in the $O_2$-sensing neurons AQR, PQR and URX (Supplementary Fig. 5) and their post-synaptic partner the RMG interneurons (Fig. 3a). *malt-1* phenotypes were rescued by expressing *malt-1* cDNA pan-neuronally, confirming that MALT-1 has neuronal functions (Fig. 1k–m). Selectively expressing *malt-1* cDNA in the RMG interneurons, or the $O_2$-sensing neurons, restored aggregation behavior to *malt-1* mutants (Fig. 3b), but only partially rescued the $O_2$-response defects (Fig. 3c). By contrast, we observed almost complete rescue of the $O_2$ response phenotype when we expressed MALT-1 simultaneously in both sets of neurons (Fig. 3c;

Supplementary Fig. 4f and g). Thus, like ILCR-1 and ILCR-2[17], MALT-1 functions in RMG and AQR, PQR and URX to promote escape from 21% $O_2$.

$Ca^{2+}$ imaging revealed that $O_2$-evoked $Ca^{2+}$ responses in RMG were significantly reduced in *malt-1* mutants, both in immobilized (Fig. 4a) and freely moving (Supplementary Fig. 4h) animals. By contrast, $O_2$-evoked $Ca^{2+}$ responses in the URX sensory neurons appeared normal in *malt-1* mutants (Fig. 4b). These phenotypes recapitulate those observed in IL-17 signaling mutants[17]. The RMG $Ca^{2+}$ response defect was rescued by expressing *malt-1* cDNA from the *npr-1* promoter, which drives expression in RMG and the AQR, PQR and URX neurons (Fig. 4a). Together, these data indicate that, like ILCR-1 and ILCR-2, MALT-1 functions in both pre-synaptic and post-synaptic neurons in the $O_2$-sensing circuit.

The *malt-1* and *ilc-17.1* mutant phenotypes were not additive. Both the $Ca^{2+}$ signaling (Fig. 4c) and behavioral response (Fig. 4d) defects of *malt-1; ilc-17.1* double mutants resembled those of single mutants, suggesting MALT-1 and ILC-17.1 function in the same pathway. Similarly, the RMG response defects of *malt-1* mutants were not enhanced by defects in PIK-1/IRAK (Supplementary Fig. 4h). Together, our biochemical, genetic, behavioral and physiological data suggest that the paracaspase MALT-1 mediates IL-17 signaling in neurons, most likely via a signaling complex made up of ACTL-1–IRAK/PIK-1–MALT-1–NFKI-1.

To examine if *malt-1* is required developmentally, we expressed it selectively in adults using a heat-shock-inducible promoter. Without heat-shock, the *phsp-16::malt-1* cDNA transgene did not

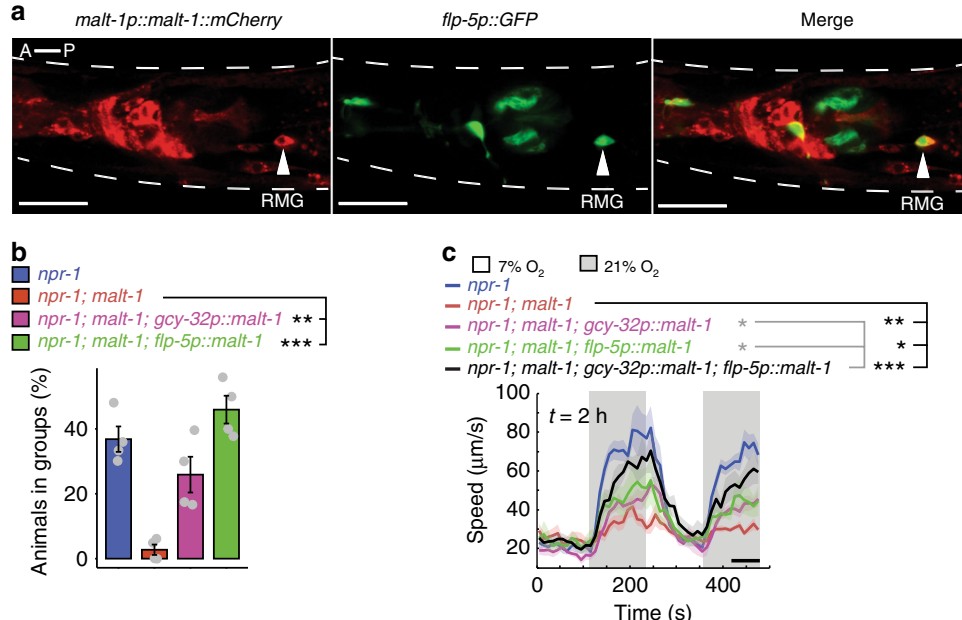

**Fig. 3 MALT-1 functions in RMG interneurons. a** A MALT-1::mCherry translational fusion, expressed from its endogenous promoter (4 kb), is expressed in RMG interneurons. RMG is recognized by its characteristic shape, location, and using a *flp-5p::gfp* reporter. Similar results were obtained in 3 experiments. Scale bars: 20 μm. **b** Expressing *malt-1* cDNA from either the *flp-5* promoter (RMG, ASG, PVT, I4, M4, and pharyngeal muscle), or the *gcy-32* promoter (URX, AQR and PQR) rescues the aggregation defect of *malt-1* mutants. N = 4 assays. Data are presented as mean values +/− SEM. **P < 0.01, ***P < 0.001, one-way ANOVA with Tukey's post hoc HSD. **c** The $O_2$-response defect of *malt-1* mutants is partially rescued by expressing *malt-1* cDNA from the *flp-5* promoter (RMG, ASG, PVT, I4, M4, and pharyngeal muscle), or the *gcy-32* promoter (URX, AQR and PQR), and almost completely rescued when *malt-1* is expressed from both promoters simultaneously. Lines indicate average speed and shaded regions indicate SEM. n = 55 animals (*npr-1*), n = 85 animals (*npr-1; malt-1*), n = 58 animals (*npr-1; malt-1; gcy-32p::malt-1*), n = 66 animals (*npr-1; malt-1; flp-5p::malt-1*), n = 46 animals (*npr-1; malt-1; gcy-32p:: malt-1, flp-5p::malt-1*). Plots show average speed (line) and SEM (shaded regions). *P < 0.05, **P < 0.01, ***P < 0.001, two-sided Mann-Whitney U test.

rescue the $O_2$-response phenotype of *malt-1* mutants (Fig. 4e). Heat-shock-induced expression during the 4th larval stage was sufficient to restore behavioral responses (Fig. 4f), suggesting that MALT-1, like other IL-17 signaling components[17], can alter circuit properties after the circuits have developed.

**MALT-1 functions as a protease in the nervous system.** In the mammalian immune system MALT1 functions both as a scaffold and as a protease. To examine if MALT-1 acts as a protease in neurons we edited the active site cysteine of the endogeneous *malt-1* gene to alanine. The equivalent mutation is used in a paracaspase-dead model in mice[38–40]. *malt-1 C374A* animals resembled *malt-1* null mutants, and could be rescued by pan-neuronal expression of *malt-1* cDNA (Fig. 5a; Supplementary Fig. 6a). By contrast, a *malt-1 C374A* transgene was unable to rescue the phenotype of *malt-1(db1194)* mutants (Fig. 5b). Unexpectedly, overexpressing *malt-1 C374A* in a WT background conferred a *malt-1(null)* phenotype (Fig. 5c), suggesting that catalytically dead MALT-1 can act as a dominant negative. Together these data suggest that MALT-1 protease activity is important for its function in the *C. elegans* nervous system.

We also asked if IL-17 signaling requires PIK-1/IRAK kinase activity. We created a single copy transgene in which the ATP-binding pocket lysine residue (K217) of PIK-1 was mutated to alanine. The *K217A* transgene rescued *pik-1(null)* phenotypes (Supplementary Fig. 6b), suggesting that kinase activity is not essential for PIK-1 to regulate behavior.

**MALT-1 promotes assembly of IL-17 signaling complexes.** To extend our in vivo proteomic analyses we made a strain in which

endogenous ACTL-1, PIK-1, MALT-1, and NFKI-1 were each tagged with different epitopes. To corroborate our LC-MS/MS data we first showed that ACTL-1, PIK-1, and NFKI-1 specifically co-immunoprecipitated with MALT-1 in a multiple knock-in strain (Fig. 6a).

To analyze the signaling complex further we carried out IPs from strains overexpressing NFKI-1-GFP. When we quantitatively compared NFKI-1 complexes from WT, *malt-1* and *pik-1* mutants, using IP/MS, we found that the amount of PIK-1/IRAK co-precipitating with NFKI-1 was reduced when MALT-1 was absent (Fig. 6b). By contrast, in *pik-1* mutants the interaction between MALT-1 and NFKI-1 was not significantly reduced (Fig. 6c). These data suggest that NFKI-1 recruitment to the signaling complex requires MALT-1.

To ask if MALT-1 and NFKI-1 interact directly, we expressed epitope-tagged versions of the proteins in *E. coli*, and performed pairwise tests for co-immunoprecipitation. MALT-1-HA immunoprecipitated NFKI-1-V5, and conversely NFKI-1-V5 immunoprecipitated MALT-1, supporting a direct physical interaction (Fig. 6d). MALT-1 also interacted directly with ACTL-1 (Fig. 6e).

Sub-domains of NFKI-1 and MALT-1 did not express well in *E. coli*. We therefore used the yeast two-hybrid assay to map domains mediating the interaction between MALT-1 and NFKI-1. We found that the DD of MALT-1 could interact with the N-terminal half of NFKI-1 (Fig. 6f), suggesting that MALT-1's DD contributes to NFKI-1 binding.

**Sub-cellular localization of IL-17 signaling components.** In the mammalian immune system IRAKs and MALT1 are core components of the Myddosome and CBM signalosome, respectively. These complexes are structurally related filamentous oligomers that assemble in the cytosol[41,42]. IκB family proteins can perform

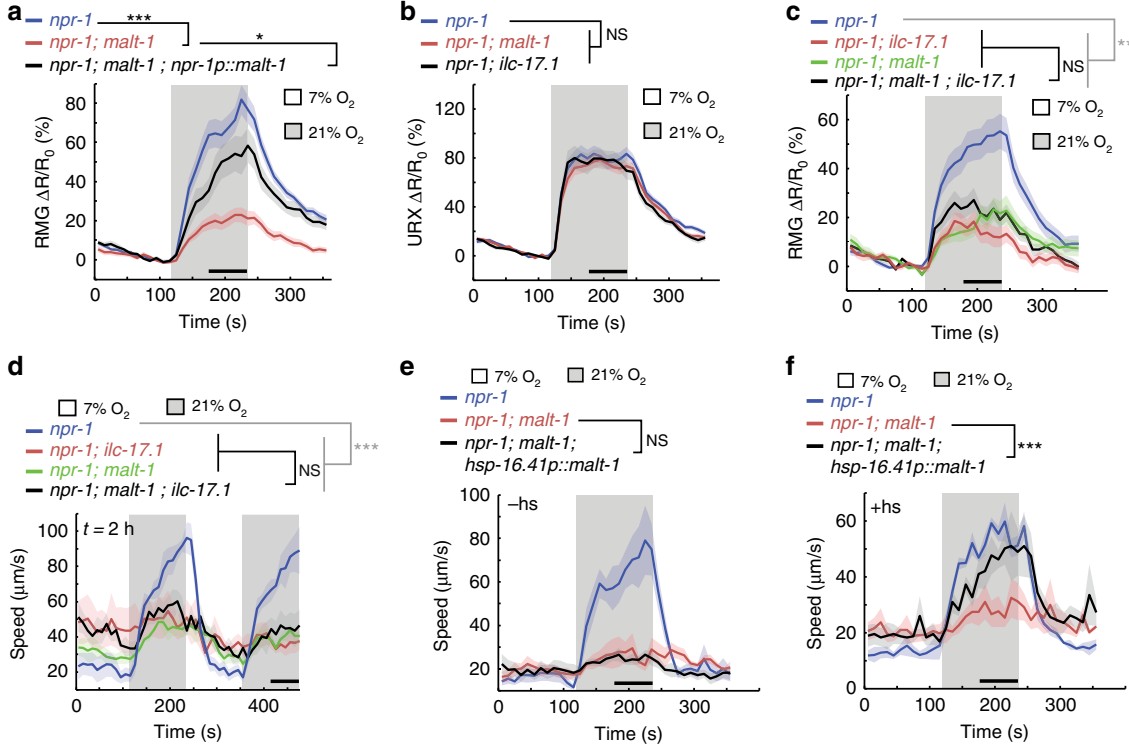

**Fig. 4 MALT-1 mediates IL-17 signaling. a** and **b** Disrupting *malt-1* attenuates Ca$^{2+}$ responses evoked by 21% O$_2$ in RMG (**a**) but not URX (**b**) neurons. The RMG defect can be rescued by expressing *malt-1* cDNA in both RMG and URX, using the *npr-1* promoter (**a**). **a** $n = 25$ animals (*npr-1*), $n = 35$ animals (*npr-1; malt-1*), $n = 24$ animals (*npr-1; malt-1; npr-1p::malt-1*). **b** n = 20 animals (*npr-1*), n = 21 animals (*npr-1; malt-1*), n = 13 animals (*npr-1; ilc-17.1*). Ca$^{2+}$ responses are reported by YC2.60 cameleon. Lines indicate average speed and shaded regions indicate SEM. $n = 20$ animals (*npr-1*), $n = 21$ animals (*npr-1; malt-1*), $n = 13$ animals (*npr-1; ilc-17.1*). *$P = 0.03$, ***$P = 0.0003$, two-sided Mann-Whitney $U$ test. **c** and **d** Null mutations in *malt-1* and *ilc-17.1* do not show additive phenotypes when either RMG Ca$^{2+}$ transients (**c**) or speed responses evoked by 21% O$_2$ are measured (**d**). Lines indicate average speed and shaded regions indicate SEM. **c** $n = 29$ animals (*npr-1*), $n = 25$ animals (*npr-1; malt-1*), $n = 28$ animals (*npr-1; ilc-17.1*), $n = 31$ animals (*npr-1; malt-1; ilc-17.1*). **d** $n = 50$ animals (*npr-1*), $n = 61$ animals (*npr-1; malt-1*), $n = 67$ animals (*npr-1; ilc-17.1*), $n = 52$ animals (*npr-1; malt-1; ilc-17.1*). **$P < 0.01$, ***$P < 0.001$, two-sided Mann-Whitney $U$ test. **e** A transgene expressing *malt-1* cDNA from the *hsp-16.41* promoter does not rescue *malt-1* phenotypes in the absence of heat-shock. Plots show average speed (line) and SEM (shaded regions). $n = 35$ animals (*npr-1*), $n = 59$ animals (*npr-1; malt-1*), $n = 47$ animals (*npr-1; malt-1; hsp-16.41p::malt-1*). $P = 0.14$, two-sided Mann-Whitney $U$ test. **f** Heat-shock-induced cDNA expression in adults restores O$_2$-evoked responses to *malt-1* mutants. Plots show average speed (line) and SEM (shaded regions); $n = 36$ animals (*npr-1*), $n = 48$ animals (*npr-1; malt-1*), $n = 33$ animals (*npr-1; malt-1; hsp-16.41p::malt-1*). Plots show average speed (line) and SEM (shaded regions). ***$P = 8.77e{-}06$, two-sided Mann-Whitney $U$ test.

both cytoplasmic and nuclear functions downstream of signalosome assembly[43]. Fractionation of a *C. elegans* lysate by gel filtration revealed that ACTL-1 and PIK-1 exist mostly as high-molecular weight species; they eluted in the heaviest fractions, including the void, of a gel filtration column (Fig. 6g and Supplementary Fig. 7). MALT-1 and NFKI-1 ran mostly as smaller species (~50–200 kDa), but they were also detectable in the heavier ACTL-1-containing and PIK-1-containing fractions. The high-molecular weight species we observed may be an artifact of unsolubilized membrane or protein aggregation, or may represent interactions with additional proteins. Alternatively, they may report oligomeric complexes of ACTL-1/PIK-1/MALT-1 related to the Myddosome and the CBM signalosome[41,42], although this hypothesis requires further testing.

To determine the sub-cellular localization of IL-17 signaling components, we separated the nuclear and cytosolic fractions of our lysate. ACTL-1-FLAG and MALT-1-HA were consistently detected in both cytoplasmic and nuclear fractions (Fig. 7a). NFKI-1-V5 was predominantly in nuclear fractions (Fig. 7a; five replicates), although as NFKI-1-V5 immunoreactivity in the fractions was weak we cannot rule out the possibility that NFKI-1 was also present in the cytoplasmic fractions at levels below our detection threshold. It is notable that NFKI-1 specifically co-immunoprecipitated with transcription factors and chromatin

state modifiers, including CREB binding protein (CBP), a histone acetyltransferase[44], suggesting that NFKI-1 regulates transcription (Supplementary Data 1c).

**MALT-1 and NFKI-1 provide partially parallel IL-17 outputs**. Overexpressing NFKI-1 suppresses *ilcr-1*, *actl-1* and *pik-1* null phenotypes, suggesting NFKI-1 functions downstream of those signaling components[17]. Overexpressing MALT-1 also rescued the O$_2$ arousal defects of *ilc-17.1*, *ilcr-1 actl-1,* and *pik-1* mutants (Fig. 7b and c). To test whether MALT-1 functions upstream or downstream of NFKI-1, we asked whether overexpressing either component rescued a null mutant of the other. Overexpressing NFKI-1 in *malt-1(null)* mutants, or MALT-1 in *nfki-1(null)* animals, fully rescued the aggregation defect but either did not restore, or only partly restored, the arousal response to 21% O$_2$ (Fig. 7d–g). These data suggest MALT-1 and NFKI-1 provide partially parallel outputs for IL-17 signaling.

**Disrupting IL-17 signaling reprograms gene expression**. In mammalian tissues IL-17 acts globally to drive pro-inflammatory gene expression[45]. We defined a transcriptional fingerprint of *C. elegans* IL-17 signaling by comparing the whole-animal RNA-seq profiles of *ilc-17.1*, *malt-1*, and *nfki-1* mutants to that of controls

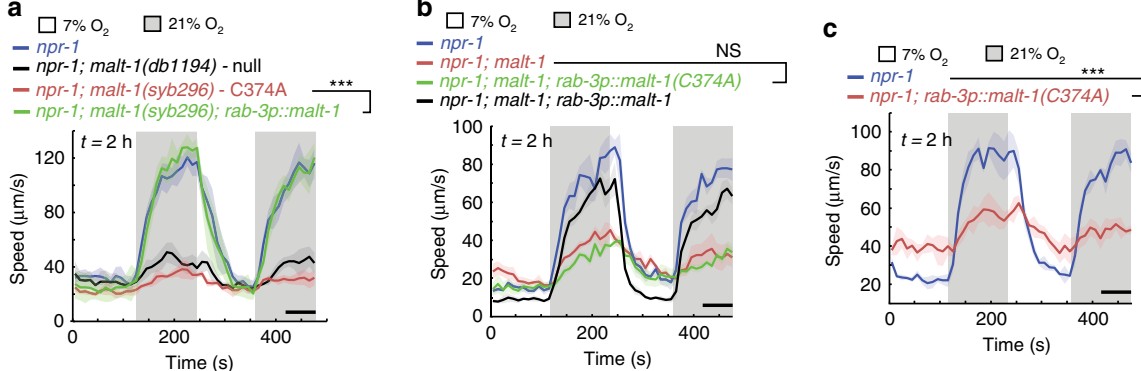

**Fig. 5 MALT-1 has enzymatic roles in IL-17 signaling. a–c** MALT-1's function in the nervous system requires its protease active site. **a** *malt-1(syb296)* mutants that express a catalytically inactive MALT-1 (C374A) show $O_2$ response defects comparable to those of *malt-1(null)* animals. Pan-neuronal expression of *malt-1* cDNA rescues this phenotype. *n* = 53 animals (*npr-1*), *n* = 55 animals (*npr-1; malt-1(db1194)*), *n* = 50 animals (*npr-1; malt-1(syb296)*), *n* = 29 animals (*npr-1; malt-1(syb296); rab-3p::malt-1*). Plots show average speed (line) and SEM (shaded regions). ***P = 2.95e−09, two-sided Mann-Whitney *U* test. **b** cDNA encoding a MALT-1 C374A catalytically inactive protein, expressed from the *rab-3* promoter, does not rescue the $O_2$ response defects of *malt-1* mutants. Data corresponding to *npr-1* and *npr-1; malt-1; rab-3p::malt-1* in **b** are the same as those shown in Fig. 1m, and were obtained in parallel to the genotypes shown. *n* = 46 animals (*npr-1*), *n* = 74 animals (*npr-1; malt-1*), *n* = 71 animals (*npr-1; malt-1; rab-3p::malt-1(C374A)*), *n* = 46 animals (*npr-1; malt-1; rab-3p::malt-1*). Plots show average speed (line) and SEM (shaded regions). NS, P = 0.693918, two-sided Mann-Whitney *U* test. **c** Overexpressing MALT-1 C374A cDNA in *npr-1* animals inhibits the arousal response to 21% $O_2$. *n* = 53 animals (*npr-1*), *n* = 87 animals (*npr-1; rab-3p::malt-1 (C374A)*). Plots show average speed (line) and SEM (shaded regions). ***P = 1.09e−12, two-sided Mann-Whitney *U* test. See also Supplementary Fig. 6.

(Supplementary Data 3). Data analysis suggested that pathways implicated in neuropeptide signaling, metabolism, ageing, and immunity were significantly altered by IL-17 signaling (Fig. 8a and b, and Supplementary Data 3).

To extend our analysis, we compared our dataset to a previous study that identified genes differentially expressed in animals acclimated to 21% and 7% $O_2$[31]. Most of the neuropeptides regulated by IL-17 were not regulated by $O_2$ experience[31] (Supplementary Data 4), suggesting IL-17 elicits transcriptional changes not explained by altered activity of the $O_2$-sensing circuit. These data suggest that IL-17 signaling may directly or indirectly alter many features of *C. elegans* behavior and global physiology.

**MALT-1 and IL-17 signaling regulate multiple behaviors.** The widespread expression of MALT-1 and other IL-17 signaling components in the nervous system, together with our RNA Seq data, suggested that IL-17 signaling forms an important neuro-modulatory axis in *C. elegans*. To begin probing this hypothesis we tested mutants in an associative learning paradigm. In this assay animals associate an environment high in NaCl with food withdrawal, which leads them to suppress salt attraction when subsequently tested in a chemotaxis assay[46]. Mutants in *ilc-17.1*, *pik-1*, and *nfki-1* exhibit normal naive responses to salt[17]. By contrast, all IL-17 signaling mutants we tested retained stronger attraction to salt than controls after conditioning (Fig. 8c). IL-17 and MALT-1 therefore regulate associative learning, as well as escape from 21% $O_2$. We could rescue the *ilcr-1* learning phenotype by selectively expressing cDNA encoding the ILCR-1 receptor in the ASE salt-sensing neurons (*flp-6p*), but not in the RMG interneurons (*flp-5p*) or the $O_2$ sensors (*gcy-32p*) (Fig. 8d), indicating that IL-17 signaling in the nervous system is not restricted to the $O_2$-sensing circuit.

**Neural IL-17–MALT-1 signaling alters immunity and lifespan.** We next assessed the impact of MALT-1 signaling on physiological phenotypes known to be regulated by the nervous system. To explore immune functions, we measured survival on *Pseudomonas aeruginosa*, a bacterial pathogen that colonizes the intestine of *C. elegans*[47]. We carried out these experiments in animals having the N2 version of the *npr-1* neuropeptide

receptor, *npr-1* 215 V, which inhibits aggregation behavior and escape from 21% $O_2$[34,48]; this ensures differences in hyperoxia avoidance do not contribute to altered pathogen resistance. To further exclude behavioral effects, we tested survival on PA14 using both small lawn assays, in which animals are able to avoid the pathogen, and big lawn assays, in which they are not[49].

Animals lacking *malt-1*, or harboring the *malt-1* protease-dead allele *malt-1(C374A)*, were resistant to PA14 infection compared to N2 controls in both small lawn (Fig. 6e and f) and big lawn assays (Supplementary Fig. 8a and b). PA14 resistance in *malt-1 (null)* mutants was rescued by pan-neuronal expression of *malt-1* cDNA (Fig. 8e and Supplementary Fig. 8a), suggesting that MALT-1 acts in the nervous system to regulate the immune response. Like *malt-1* mutants, *ilcr-1* and *nfki-1* mutants survived significantly longer on PA14 than controls (Fig. 8g, h and Supplementary Fig. 8c, d).

Increased pathogen resistance is often associated with increased lifespan. To examine if disrupting IL-17 signaling alters lifespan we measured survival on the standard laboratory food source of *C. elegans*, *E. coli* OP50. *ilc-17.1* and *malt-1* mutants lived significantly longer than N2 controls (Fig. 8j, k). Expression of *ilc-17.1* cDNA from its endogenous promoter not only rescued the phenotype of the null mutant, but significantly reduced lifespan compared to non-transgenic N2 controls (Fig. 8k). We could rescue the extended lifespan of *malt-1* mutants by pan-neuronal expression of *malt-1* (Fig. 8j), suggesting that IL-17 signaling acts in the nervous system to regulate longevity. The lifespan phenotypes of *malt-1* and *ilc-17.1* mutants were not additive (Fig. 8l). Furthermore, the ability of ILC-17.1 overexpression to reduce lifespan was dependent on *malt-1* (Fig. 8m). Together, these two observations suggest that MALT-1 acts downstream of ILC-17.1 to negatively regulate longevity.

MALT-1 strongly and specifically co-immunoprecipitated with factors known to regulate longevity or immunity, including NHR-49[50] and TIR-1[26,27] (Supplementary Data 1d). TIR-1 (Toll/Interleukin-1 Receptor domain protein), the *C. elegans* ortholog of SARM1 (Sterile alpha and TIR motif containing protein), functions upstream of the p38 MAPK pathway[51] to upregulate expression of anti-microbial peptides, including the ShK-like toxin T24B8.5 in the intestine[52,53] and immune responses to *P. aeruginosa*[27]. Like *tir-1*

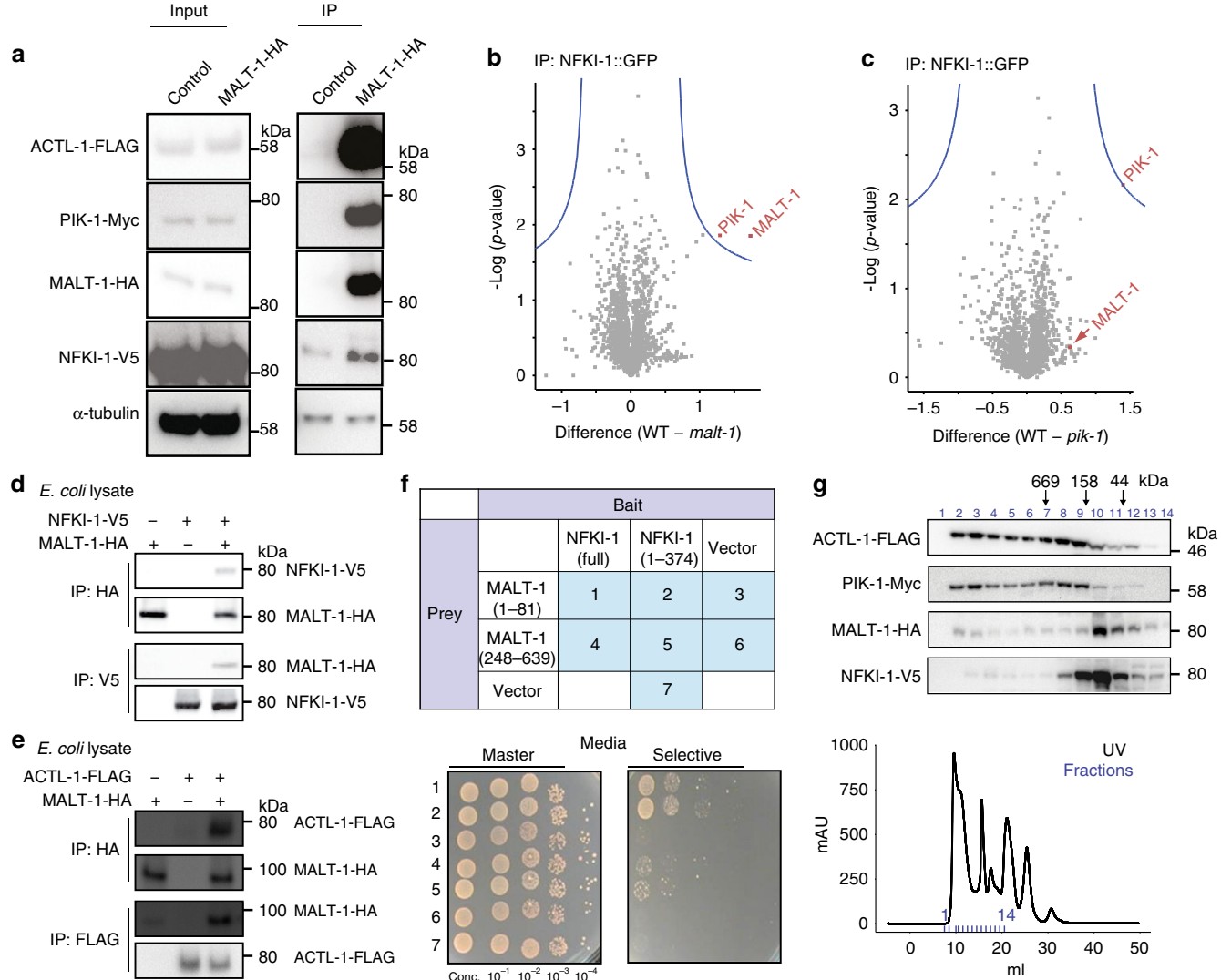

**Fig. 6 MALT-1 has scaffolding roles in IL-17 signaling. a** Endogenous ACTL-1, PIK-1 and NFKI-1 co-IP with endogenous MALT-1 in *npr-1* animals. Anti-HA antibody was used to immunoprecipitate MALT-1 complexes. Half of the lysate was immunoprecipitated with anti-IgG as a control. Tags were knocked in by CRISPR. Similar results were obtained in 3 experiments. **b** and **c** Volcano plot showing quantitative LC-MS/MS of proteins that interact with NFKI-1::GFP in *malt-1* and *pik-1* mutants compared to wild type. NFKI-1::GFP was purified using GFP-Trap beads, and immunoprecipitated proteins labeled using tandem mass tags (TMT-labeling). The average relative abundance in two biological replicates is shown. *p*-values are reported by a two sample *t*-test. The amount of PIK-1 that co-IPs with overexpressed NFKI-1::GFP is significantly reduced in *malt-1(db1194)* mutants (**b**). The relative amount of MALT-1 that co-IPs with NFKI-1 is not significantly decreased in *pik-1(tm2167)* mutants (**c**). Peptides derived from MALT-1 and PIK-1 are shown in Supplementary Data 2. **d** and **e** IPs of His10-tagged *C. elegans* ACTL-1-FLAG, MALT-1-HA, and NFKI-1-V5 recombinantly expressed in *E. coli* show that MALT-1 can directly bind NFKI-1 (**d**) and ACTL-1 (**e**). **d** was performed once, e was performed three times with similar results. **f** Interaction of the MALT-1 Death Domain (1-81) with the N-terminus of NFKI-1 (1-374) in a yeast two-hybrid assay using nutritional selection (*ADE2*). Rows show 10-fold serial dilutions of each of the seven Prey–Bait combination strains tested and shown top. Similar results were obtained in 2 experiments. **g** Elution profiles of ACTL-1, PIK-1, MALT-1, and NFKI-1 proteins in a *C. elegans* extract run on a Superose 6 Gel Filtration column and visualized by immunoblot. All four proteins can be found in high molecular weight complexes. Similar profiles were observed in two runs. See also Supplementary Fig. 7 and Supplementary Data 2.

mutants[53], *malt-1* and *ilcr-1* mutants showed reduced T24B8.5 expression (Supplementary Fig. 9a), and this reduction could be rescued by either intestine-specific or nervous system-specific expression of *malt-1* (Supplementary Fig. 9b). However, PA14 resistance was reduced in *malt-1; tir-1* double mutants compared to *malt-1* (Fig. 8i). Thus TIR-1/SARM can still promote PA14 resistance in *malt-1* mutants, and while overall IL-17 signaling inhibits the *C. elegans* immune response to PA14, this effect may reflect the net outcome of opposing influences.

In summary, our data suggest that IL-17 signals through a MALT-1 signalosome to modify neural properties and remodel the behavior and physiology of *C. elegans* (Fig. 9).

## Discussion

Our data suggest that MALT1 modulates neural circuit function in *C. elegans*, by acting as a protease and a scaffold. MALT-1 participates in an ACTL-1-IRAK-MALT-1 signaling complex that mediates IL-17 signaling. The high molecular weight of this complex in *C. elegans* extracts suggests it may form a structure related to the MYD88-IRAK4-IRAK2 Myddosome[41] and CARMA1-BCL10-MALT1 CBM signalosome[42], although this hypothesis needs further study. MALT-1 directly binds ACTL-1 in vitro, and yeast two hybrid data suggest ACTL-1 directly binds *C. elegans* IRAK[54]. MALT-1 also interacts directly with NFKI-1, a homolog of mammalian IκBζ/IκBNS, and can signal through both NFKI-1-dependent

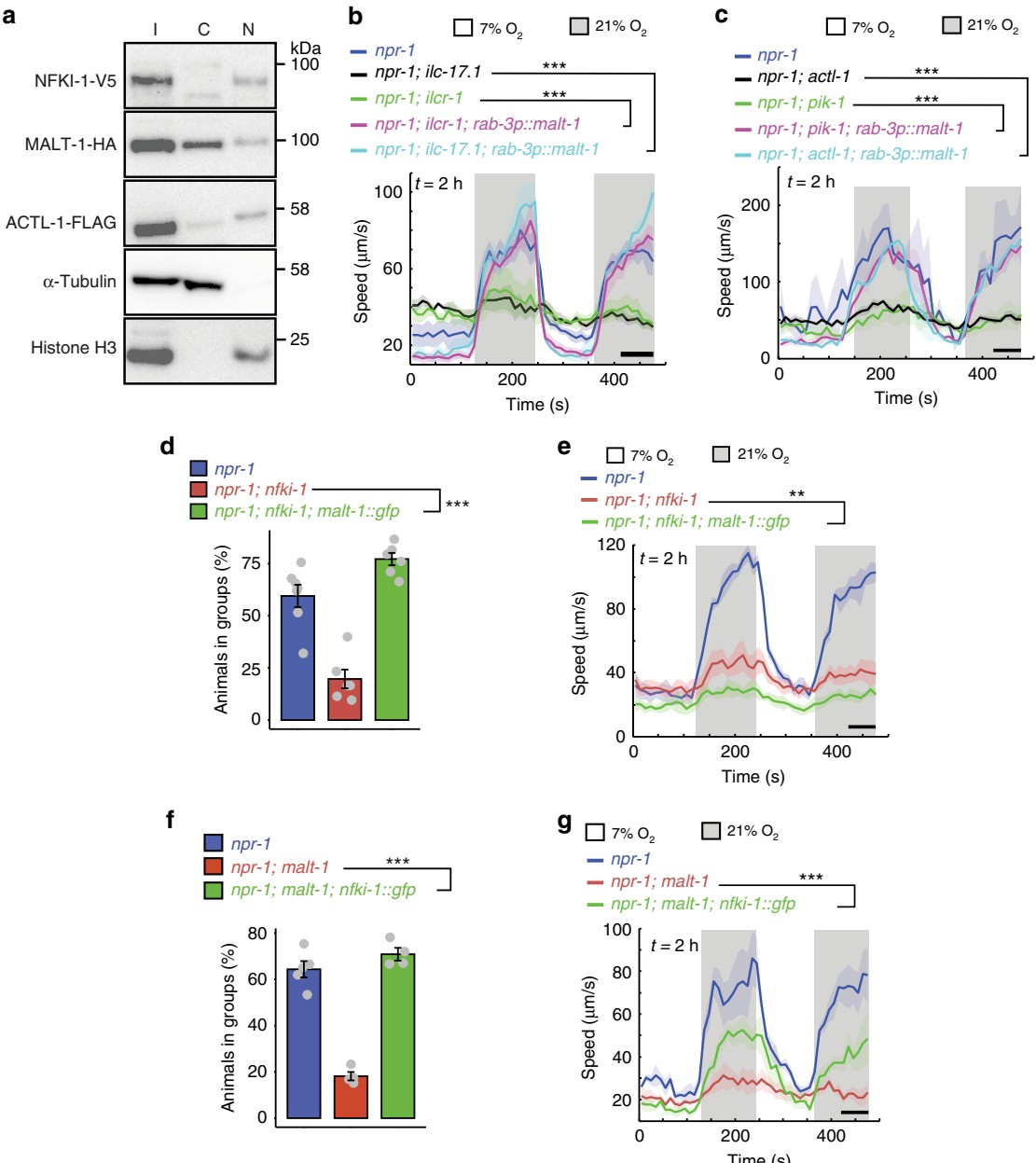

**Fig. 7 MALT-1 and NFKI-1 provide partially parallel outputs of IL-17 signaling. a** Immunoblot analysis of IL-17 signaling components from nuclear and cytoplasmic fractions of *C. elegans* lysate. I, input, C, cytosolic, N, nuclear. NFKI-1 is predominately nuclear; ACTL-1 and MALT-1 are distributed between the nucleus and cytoplasm. Similar results were obtained in 5 experiments. **b** and **c** Overexpressing *malt-1* in neurons, using the *rab-3* promoter, restores the arousal response to 21% O$_2$ to *ilc-17.1* and *ilcr-1* mutants (**b**), and *actl-1* and *pik-1* mutants (**c**). **b** n = 52 animals (*npr-1*), n = 104 animals (*npr-1; ilcr-1*), n = 71 animals (*npr-1; ilcr-1; rab-3p::malt-1*), n = 86 animals (*npr-1; ilc-17.1*), n = 61 animals (*npr-1; ilc-17.1; rab-3p::malt-1*). **c** n = 19 animals (*npr-1*), n = 46 animals (*npr-1; actl-1*), n = 26 animals (*npr-1; actl-1; rab-3p::malt-1*), n = 33 animals (*npr-1; pik-1*), n = 28 animals (*npr-1; pik-1; rab-3p::malt-1*). Plots show average speed (line) and SEM (shaded regions). ***P < 0.001, two-sided Mann-Whitney U test. **d** and **e** Overexpressing *malt-1* gDNA also rescues the aggregation phenotype (**d**), but not the arousal defect (**e**) of *nfki-1* mutants. **d** N = 7 assays (*npr-1*), N = 6 assays (*npr-1; nfki-1* and *npr-1; nfki-1; malt-1::gfp*). ***P = 3.5e−05, one-way ANOVA with Tukey's post hoc HSD. **e** n = 47 animals (*npr-1*), n = 79 animals (*npr-1; nfki-1*), n = 39 animals (*npr-1; nfki-1; malt-1::gfp*). **P = 0.0067, two-sided Mann-Whitney U test. **f** and **g** The aggregation phenotype of *malt-1* is rescued by overexpressing *nfki-1* cDNA (**f**), while speed defects are partially rescued (**g**). **f** N = 5 assays (*npr-1*), N = 4 assays (*npr-1; malt-1* and *npr-1; malt-1; nfki-1::gfp*). ***P = 7e−07, one-way ANOVA with Tukey's post hoc HSD. **g** n = 36 animals (*npr-1*), n = 50 animals (*npr-1; malt-1*), n = 44 animals (*npr-1; malt-1; nfki-1::gfp*). ***P = 7e−07, one-way ANOVA with Tukey's post hoc HSD. ***P = 4.7e−4, two-sided Mann-Whitney U test.

and independent mechanisms to alter neuron function and change behavior. The ACTL-1-IRAK-MALT-1-NFKI-1 pathway is present in most neurons of the *C. elegans* nervous system, and appears to be a neuromodulatory axis impacting multiple phenotypes.

Like ILCR-1 and ILCR-2[17], MALT-1 functions in both URX O$_2$ sensors and RMG interneurons to promote escape from 21%

O$_2$. In RMG, ILC-17.1/MALT-1 signaling potentiates Ca$^{2+}$ responses to pre-synaptic input from URX O$_2$ sensors, which are tonically activated by 21% O$_2$. In URX, ILC-17.1/MALT-1 signaling does not appear to disrupt O$_2$-evoked Ca$^{2+}$ responses, suggesting that it potentiates behavioral arousal to 21% O$_2$ by augmenting synaptic or gap junctional communication. These

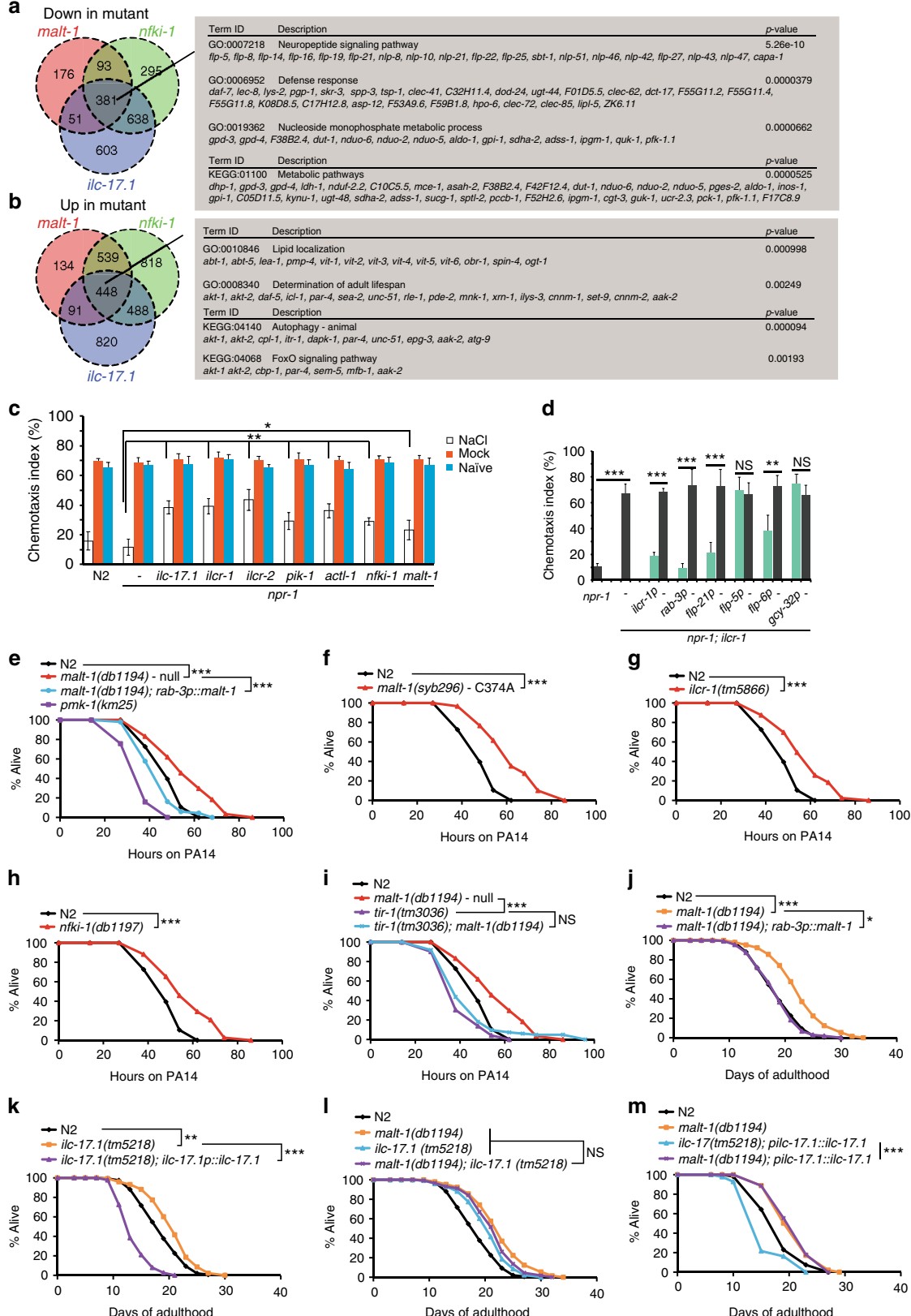

different effects of IL-17 signaling may be indicative of cell-type specific effects on gene expression. Our IP/MS experiments identified transcription factors, chromatin remodeling factors and RNA binding proteins as specific interactors of NFKI-1 and/or MALT-1, but further work is needed to identify cell types in which these interactions are functionally relevant.

Neuronal MALT-1 signaling also modulates pathogen susceptibility and longevity. The nervous system plays an important and conserved role in regulating immunity[55–59], and multiple neurons[60–62] and secreted factors[63,64] that regulate innate immune gene expression in non-neuronal tissues have been discovered. The nervous system also mediates behavioral avoidance of pathogens, by

**Fig. 8 MALT-1 acts downstream of IL-17 signaling to reprogram behavior and physiology. a** and **b** Downregulated (**a**) and upregulated (**b**) genes in whole animal RNA-seq profiles of *malt-1; npr-1, ilc-17.1; npr-1* and *nfki-1; npr-1* double mutants compared to *npr-1* controls. Gene ontology (GO) terms and KEGG pathways significantly overrepresented among genes dysregulated in all three mutant conditions are shown (*q*-value <0.05, with a minimum log2 (fold-change) of ±0.25). **c** and **d** Salt chemotaxis after conditioning by food-withdrawal in the absence or presence of NaCl. *$P < 0.05$, **$P < 0.01$, ***$P < 0.001$, one-way ANOVA with Tukey's post hoc HSD, $N = 6$ assays. **d** The salt chemotaxis learning defect of *ilcr-1* mutants is rescued by driving *ilcr-1* expression in many neurons (*rab-3* or *flp-21* promoters), or specifically in ASE (*flp-6* promoter). **e–i** PA14 big lawn assays. $n \geq 81$ animals. ***$P < 0.001$, two-sided logrank test; precise n numbers and *P* values are provided in Supplementary Table 1. Mutants lacking *malt-1* (**e**) or encoding protease-dead *malt-1* (**f**), or defective in other IL-17 signaling components (**g, h**) are resistant to *P. aeruginosa* PA14 in big lawn assays, where animals cannot escape from the PA14 lawn. The enhanced survival of *malt-1* mutants is rescued by pan-neuronal expression of *malt-1* gDNA. $n \geq 81$ animals. **i** The enhanced resistance of *malt-1* mutants to PA14 requires TIR-1. Like *tir-1* mutants, *malt-1; tir-1* double mutants are hypersensitive to PA14 infection. **j–m** Lifespan. $n \geq 92$ animals. **$P < 0.01$, ***$P < 0.001$, two-sided logrank test; precise n numbers and *P* values are provided in Supplementary Table 3. **j** and **k** The lifespan of *malt-1* and *ilc-17.1* mutants is increased compared to N2 controls. The *malt-1* phenotype is rescued by expressing *malt-1* gDNA pan-neuronally (**j**) and the *ilc-17.1* phenotype can be rescued by expressing *ilc-17.1* cDNA from its endogenous promoter (**k**). **l** The lifespan phenotypes of *malt-1* and *ilc-17.1* mutants are not additive. **m** The shortened lifespan of animals overexpressing ILC-17.1 is abolished in *malt-1* mutants. See also Supplementary Fig. 7, Supplementary Tables 1–3 and Supplementary Data 3–6.

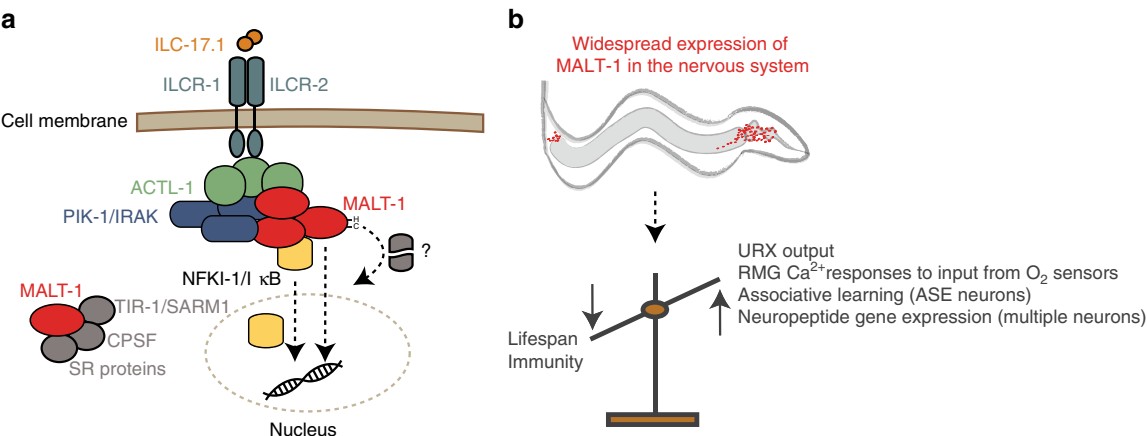

**Fig. 9 Model. a** Activation of nematode IL-17Rs ILCR-1 and ILCR-2 engages ACTL-1, the *C. elegans* ACT1-like adapter, probably via their SEFIR domains. ACTL-1 recruits the *C. elegans* IRAK and MALT1 homologs to form the ACT1-IRAK-MALT1 signalosome in the cytoplasm. This serves a scaffolding function to recruit IκBζ/NFKI-1, and modulate its actvity by an unknown mechanism. NFKI-1 probably orchestrates changes in the transcriptome of RMG and other cells. MALT1-mediated cleavage of unknown substrate(s) positively regulates NFKI-1 signaling. In parallel to this pathway, MALT-1 forms a complex of unknown function with TIR-1/SARM1, and with multiple RNA-binding proteins. **b** ILCR receptors and downstream signaling components including MALT-1 are expressed in many neurons. This neuronal signaling cassette alters associative learning, as well as O₂-escape behaviors, and suppresses lifespan and immunity.

mechanisms that can be innate or learned[49,65]. Our data suggest that neuronal ILC-17.1/MALT-1 signaling reduces survival on *Pseudomonas aeruginosa* by non-behavioral mechanisms. A simple model is that by altering neural circuit activity ILC-17.1 can change immune gene expression, for example in the intestine.

MALT1-like paracaspases are found in organisms lacking other CBM components[18], suggesting MALT1 has unknown functions that predate its coaction with Bcl10 and CARD domain proteins. Our results raise the possibility that one ancestral function was in IL-17 signaling. As IL-17Rs are found throughout metazoa[66], we speculate that the ACTL-1-IRAK-MALT-1 complex we have identified is the original and primary mechanism by which IL-17Rs signal in non-amniote animals, from cnidarians to cephalochordates. In amniotes, ACT1 orthologs have lost a death domain (DD) that is present in ACT1 orthologs from most other lineages[66]. DDs mediate homotypic interactions in large immune complexes such as the Myddosome[67], and are present in both MALT1 and IRAKs. The DD–SEFIR domain architecture of ACT1 present in non-amniotes resembles the DD–TIR domain structure of MyD88, since TIR and SEFIR domains are related[68]. Interestingly, proximity labeling studies find MALT1 associates with MyD88 in DLBCL cells[69], although the functional consequences of this interaction are not yet known. Recent studies have also speculated that mammalian MALT1 is recruited to IL-

17 signaling complexes[70,71]. Direct evidence for this is lacking, but if correct, this would mirror our results in *C. elegans*.

One area of future study will investigate how MALT-1 alters neural function. Although we find that MALT-1 protease activity is essential, we have not identified its neural substrate(s). Substrates of mammalian MALT1 with orthologs in *C. elegans* include the RNA binding proteins roquin-1/2 (RLE-1), which target RNAs for degradation, the endoribonuclease regnase-1 (REGE-1), and the CYLD (CYLD-1) deubiquitinase[72–74]. We did not detect these proteins in our proteomic analyses (Supplementary Data 1d). However, our IP/MS data indicate that besides binding NFKI-1, MALT-1 interacts with multiple RNA binding proteins, including splicing and polyadenylation factors, and with the *C. elegans* ortholog of SARM1, called TIR-1. TIR-1 regulates *C. elegans* gustatory and olfactory plasticity[29], and proteostasis[75], by modulating MAPK pathways, making it a plausible target for regulation by MALT-1.

The closest mammalian homolog of NFKI-1, IκBζ, is a nuclear-localized protein that acts as a transcriptional regulator, and is rapidly induced by inflammatory stimuli, including IL-17. IκBζ is thought to mediate its effects on gene expression primarily by regulating chromatin structure, although how it is recruited to target genes is not completely understood since it lacks a DNA binding domain[76,77]. Our IP/MS data find NFKI-1 physically

interacts with the CREB binding protein (CREBBP), *cbp-1*, which is a histone acetyltransferase, consistent with NFKI-1 acting to modify chromatin structure. Most of the gene expression changes highlighted by our RNA-seq studies likely reflect secondary consequences of IL-17 signaling defects, although some genes may be directly regulated by NFKI-1.

Our biochemical and genetic analyses of IL-17 signaling in *C. elegans* have identified functional roles and biochemical interactions previously undescribed in mammals. An outstanding challenge is to examine which of these are conserved in mammals. Does MALT1 play a role in modulating mammalian neural function, given that neurons can express receptors with an ITAM (immunoreceptor tyrosine-based activation motif)[19–21], as well as GPCRs, that in immune cells signal through MALT1? Does MALT1 contribute to known neuronal responses to IL-17 [13,14,78]? Does mammalian MALT1 physically interact with IκBζ, IκBNS, or SARM1? In summary, we have used an invertebrate model system, with the relative simplicity this offers, and its advantages for genetics, biochemistry and single neuron analysis, to probe how key immune molecules signal in neurons to alter circuit function.

## Methods

**Strains and genetics**. *C. elegans* were maintained on nematode growth medium (NGM) at room temperature (22 °C) with *E. coli* OP50 food. Strains used are provided in Supplementary Table 5.

Whole genome sequencing showed that the aggregation-defective AX3621 strain was defective in *malt-1*. We used SNP mapping[79] to investigate if the aggregation defect was linked to *malt-1*. We crossed AX3621 animals with the AX288 [*lon-2(e678) npr-1(ad609)*] strain; AX288 was constructed by backcrossing *lon-2 npr-1(ad609)* X 16× into the CB4856 (Hawaiian) wild strain. The *npr-1 (ad609)* allele confers stronger aggregation than the CB4856 Hawaiian strain routinely used for mapping. We 'singled' F2 animals, and scored their progeny for aggregation. Animals from non-aggregating F3 lines were pooled, and their DNA extracted and sequenced. Sequencing libraries were made using the Nextera DNA Library kit, and sequenced on a HiSeq 2500 (Illumina) machine with 125 bp paired-end reads. Sequencing data were analyzed using CloudMap[79].

**Molecular biology**. Primers used in this study are provided in Supplementary Table 6. *C. elegans* expression constructs were generated using MultiSite Gateway Recombination (Invitrogen). To amplify the *malt-1* promoter (4 kb) we used primers ggggACAACTTTGTATAGAAAAGTTGctgc cggtggattccaacatattg and ggggACTGCTTTTTTGTACAAACTTGtctgaaattggggttcaagaaatttattttttgattttttaaaata

to amplify the *malt-1* ORF (gDNA):
ggggACAAGTTTGTACAAAAAAGCAGGCTtttcagaaaaatgaacacaaacttggcggagt tc and ggggACCACTTTGTACAAGAAAGCTGGGTActgtagacatttgattcttgtaatcaa aatatgaccaatatc

and to amplify *malt-1* cDNA:
ggggACAAGTTTGTACAAAAAAGCAGGCTtttcagaaaaatgaacacaaacttggcggagt tc and ggggACCACTTTGTACAAGAAAGCTGGGTATTACTGTAGACATTTGA TTC TTGTAATCAAAATATGACCAATATCAACATTC.

The Q5 Site-Directed Mutagenesis Kit (NEB) was used to create *malt-1(C374A)* cDNA, with the following primers: TCTTGATGTCgcCAGAAAATTTGTTCCATATG and gcgcgtcaagttgtGCCTGAC GACGAGTTGTGCTGTTTTAGAGCTAGAA.

To generate deletions in the *malt-1* locus by CRISPR/Cas9 we expressed a gaucagguauccaccguag short guide from the *rpr-1* promoter[80]. The primers used to amplify this sequence for insertion into an *Eco*RI-cut expression plasmid (addgene #48961) were gcgcgtcaagttgtGgatcaggtatccaccgtagGTTTTAGAGCTAGAA and TTCTAGCTCTAAAACctacggtggatacctgatcCacaacttgacgcgc.

Expression constructs were injected at 50 ng/μl, with the exception of CRISPR-Cas9 mixes that were prepared as previously described[80]: 30 ng ng/μl *eft-3::cas9*, 100 ng/μl sgRNA, 30 ng ng/μl cc::GFP.

The following alleles were generated by SunyBiotech (Fuzhou, China) using CRISPR/Cas9-based genome editing: *malt-1(syb296)*, *actl-1(syb412)*, *pik-1(syb378)*, *malt-1(syb573)*, and *nfki-1(syb617)*. We verified modified sequences using Sanger sequencing (Supplementary Table 4).

**Behavioral assays**. Behavioral assays were performed at room temperature (22 °C). Aggregation was assayed as previously described[48]; 60 young adults were picked onto a plate seeded with 100 μl OP50 48 h previously. Animals were left undisturbed for 2 h and then scored blind to genotype. The % of animals in groups was calculated, with a group defined as 3 or more animals in contact with one another. Statistical comparisons were made using ANOVA (with RStudio (v 1.0.143)).

Locomotory responses to $O_2$ stimuli were measured as described previously[31,81] with minor modifications. 15–25 young adults were picked onto a plate seeded with

20 μl OP50 14–18 h previously, and covered with a microfluidic PDMS chamber. Defined $O_2$ mixtures (balance nitrogen) were bubbled through $H_2O$ and delivered to the PDMS chamber at a rate of 1.4 ml/min using a PHD 2000 Infusion syringe pump (Harvard Apparatus). Video recordings were acquired at 2 fps with FlyCapture 1.X software (FLIR Systems), using a Point Gray Grasshopper camera mounted on a Leica MZ6 microscope. Speed and reversals were measured using Zentracker custom software (https://github.com/wormtracker/zentracker). To measure phenotypes associated with IL-17 signaling defects, worms were left undisturbed for 2 h on assay plates prior to recording.

To measure thrashing, single animals were placed into individual wells containing 50 μl M9 buffer. The number of complete body bends per minute was measured by a scorer blind to genotype.

**Heat-shock**. As reported previously, the *hsp-16.41* heat shock promoter is leaky in animals grown at room temperature. We therefore kept animals at 15 °C until the time of heatshock (late L4). To induce heat-shock, parafilm-wrapped plates were submerged in a 34 °C water bath for 30 min, and then recovered at room temperature until the time of assay.

**Light microscopy**. Worms were immobilized with 25 mM sodium azide on 2% agarose pads. Z stacks from animals expressing MALT-1::GFP and MALT-1::RFP were acquired on an Inverted Leica SP8 confocal microscope using a ×63/1.20 water objective, using the LAS X software platform (Leica). Figure panels were obtained using the Z-project (average intensity) function in FIJI (ImageJ v2.0.0-rc-69).

We quantified GFP intensity in L4 animals expressing the *agIs219(pT24B8.5:: GFP)* transgene[53] using NIS-elements (Nikon) and a Nikon Ti2 microscope with a Niji LED light source (Bluebox Optics, Huntingdon, UK) and a NEO scientific CMOS camera (Andor, Belfast, UK), with a ×10 objective (Nikon, Tokyo, Japan) and 50 ms exposure time.

**Calcium imaging**. Animals expressing cameleon YC2.60 were imaged with a ×2 AZ-Plan Fluor objective (Nikon) on a Nikon AZ100 microscope fitted with ORCA-Flash4.0 digital cameras (Hamamatsu). Excitation light was provided from an Intensilight C-HGFI (Nikon), through a 438/24 nm filter and an FF458DiO2 dichroic (Semrock). Emission light was split using a TwinCam dual camera adapter (Cairn Research) and passed through CFP (483/32 nm) and YFP (542/27) filters, and a DC/T510LPXRXTUf2 dichroic. We acquired movies using NIS-Elements (Nikon), with 100 ms exposure time.

To image neural activity in freely moving animals (Supplementary Fig. 4g), single young adults were transferred to peptone-free agar plates spotted with 4 μl of concentrated OP50 food in M9 buffer, and imaged at 2× zoom. For all other figures, 4–8 young adults were transferred to peptone-free agar plates, immobilized on a 2 μl patch of concentrated OP50 in M9 buffer using Dermabond adhesive, leaving the nose exposed, and imaged at 4× zoom.

**Immunoprecipitation from *C. elegans***. For co-IP experiments analyzed by LC-MS/MS, *C. elegans* lysis and affinity purification was performed as previously described[82] with minor modification. Lysis buffer A was prepared with 50 mM HEPES (pH 7.4), 1 mM EGTA, 1 mM $MgCl_2$, 100 mM KCl, 10% glycerol, 0.05% NP40, 1 mM DTT, 0.1 M PMSF and 1 complete EDTA-free proteinase inhibitor cocktail tablet (Roche Applied Science) per 12 ml. Unsynchronized worms grown in liquid were washed twice in M9 and once in ice-cold lysis buffer A, then snap-frozen by dropwise addition to $LN_2$ in preparation for cryogenic grinding. Worm popcorn was pulverized using a Freezer/Mill (SPEX SamplePrep). Crude extract was clarified at 4 °C for 10 min at 20,000×g, and again for 20 min at 100,000×g with a TLA-100 rotor (Beckman Coulter). For IP, roughly equal volumes of sample and control lysate were incubated with 100 μl GFP-Trap MA (ChromoTek gtma), Myc-Trap MA (ChromoTek ytma), or anti-FLAG M2 magnetic beads (Sigma M8823) for 3–4 h at 4 °C, then washed twice with 50 mM HEPES, 100 mM KCl. Purified complexes were eluted in SDS-sample buffer at 95 °C and fractionated by SDS-PAGE prior to characterization by LC-MS/MS.

For co-IP experiments analyzed by Western blot, the following modifications were made. Lysis buffer B contained 50 mM HEPES (pH 7.4), 100 mM KCl, 0.05% NP40, 1 mM DTT, 0.1 M PMSF and 1 complete EDTA-free proteinase inhibitor cocktail tablet (Roche Applied Science) per 12 ml. Crude extract was clarified at 4 °C for 30 min at 18,000×g. For immunoprecipitation, half of the lysate was incubated with anti-HA agarose (Sigma A2095) for 30 min at 4 °C, then washed 3× with 50 mM HEPES, 100 mM KCl. As a control, the other half of the lysate was incubated with IgG-agarose (Sigma A0919).

**Identification of protein-protein interactions by MS**. Gel samples were destained with 50% v/v acetonitrile and 50 mM ammonium bicarbonate, reduced with 10 mM DTT, and alkylated with 55 mM iodoacetamide. Proteins were digested with 6 ng/μl trypsin (Promega, UK) overnight at 37 °C, and peptides extracted in 2% v/v formic acid 2% v/v acetonitrile, and analyzed by nano-scale capillary LC-MS/MS (Ultimate U3000 HPLC, Thermo Scientific Dionex) at a flow of ~300 nL/ min. A C18 Acclaim PepMap100 5 μm, 100 μm × 20 mm nanoViper (Thermo Scientific Dionex), trapped the peptides prior to separation on a C18 Acclaim PepMap100 3 μm, 75 μm × 250 mm nanoViper. Peptides were eluted with an

acetonitrile gradient. The analytical column outlet was interfaced via a nano-flow electrospray ionization source with a linear ion trap mass spectrometer (Orbitrap Velos, Thermo Scientific). Data dependent analysis was performed using a resolution of 30,000 for the full MS spectrum, followed by ten MS/MS spectra in the linear ion trap. MS spectra were collected over a m/z range of 300–2000. MS/MS scans were collected using a threshold energy of 35 for collision-induced dissociation. LC-MS/MS data were searched against the UniProt KB database using Mascot (Matrix Science), with a precursor tolerance of 10 ppm and a fragment ion mass tolerance of 0.8 Da. Two missed enzyme cleavages and variable modifications for oxidized methionine, carbamidomethyl cysteine, pyroglutamic acid, phosphorylated serine, threonine and tyrosine were included. MS/MS data were validated using the Scaffold program (v4.10.0, Proteome Software Inc).

**Quantification of NFKI-1 interacting peptides by TMT labeling**. Protein samples on beads were reduced with 10 mM DTT at 56 °C for 30 min and alkylated with 15 mM iodoacetamide (IAA) in the dark at 22 °C for 30 min. Alkylation was quenched by adding DTT and the samples digested with trypsin (Promega, 1.25 µg) overnight at 37 °C. After digestion, supernatants were transferred to a fresh Eppendorf tube, the beads were extracted once with 80% acetonitrile/ 0.1% TFA and combined with the corresponding supernatant. The peptide mixtures were then partially dried in a Speed Vac and desalted using home-made C18 (3 M Empore) stage tip filled with 4 µl of poros R3 (Applied Biosystems) resin. Bound peptides were eluted sequentially with 30%, 50%, and 80% acetonitrile in 0.1%TFA and lyophilized.

Dried peptide mixtures from each condition were re-suspended in 40 µl of 250 mM triethyl ammonium bicarbonate. 0.8 mg of TMT 6plex reagents (Thermo Fisher Scientific) was re-constituted in 41 µl anhydrous MeCN. Twenty microliter of TMT reagent was added to each peptide mixture and incubated for 1 h at r.t. The labeling reactions were terminated by incubation with 4.4 µl of 5% hydroxylamine for 15 min. For each condition, the labeled samples were pooled, Speed Vac to remove acetonitrile, desalted and then fractionated with home-made C18stage tip using 10 mM ammonium bicarbonate and acetonitrile gradients. Eluted fractions were acidified, partially dried down in speed vac and ready for LC-MSMS.

Peptides were separated on an Ultimate 3000 RSLC nano System (Thermo Scientific), using a binary gradient consisting of buffer A (2% MeCN, 0.1% formic acid) and buffer B (80% MeCN, 0.1% formic acid). Eluted peptides were introduced directly via a nanospray ion source into a Q Exactive Plus hybrid quardrupole-Orbitrap mass spectrometer (Thermo Fisher Scientific). The mass spectrometer was operated in standard data dependent mode, performed survey full-scan (MS, $m/z$ = 350–1600) with a resolution of 140,000, followed by MS2 acquisitions of the 15 most intense ions with a resolution of 35,000 and NCE of 33%. MS target values of 3e6 and MS2 target values of 1e5 were used. Dynamic exclusion was enabled for 40 s.

The acquired MSMS raw files were processed using MaxQuant[83] with the integrated Andromeda search engine (v.1.5.5.1). MSMS spectra were searched against the *Caenorhabditis elegans* UniProt Fasta database (July 2017). Carbamidomethylation of cysteines was set as fixed modification, while methionine oxidation and N-terminal acetylation (protein) were set as variable modifications. Protein quantification required 1 (unique + razor) peptide. Other parameters in MaxQuant were set to default values. MaxQuant output file, proteinGroups.txt was then processed with Perseus software (v 1.5.5.0). After uploading the matrix, the data was filtered, to remove identifications from reverse database, modified peptide only, and common contaminants. Each peptide channel was normalized to the median and log2 transformed.

**Protein expression in *E. coli***. His10-tagged *C. elegans* MALT-1, ACTL-1, and NFKI-1 were expressed in *E. coli* strain BL21(DE3) and purified using Ni-NTA agarose (Qiagen) or HisPur Cobalt resin (ThermoFisher Scientific).

**Sub-cellular fractionation**. Nuclear/cytoplasmic fractionation was performed as described previously[84]. Young adult worms were washed 3–5 times in M9, and twice in hypotonic buffer (15 mM HEPES, 10 mM KCl, 5 mM MgCl₂, 0.1 mM EDTA, 350 mM sucrose). Lysis was on ice in complete hypotonic buffer plus 1 mM DTT and 1 complete EDTA-free proteinase inhibitor cocktail tablet (Roche Applied Science) per 12 ml, using a motorized pellet pestle (Sigma Z359971, Z359947) until most worm carcasses were homogenized. Worm debris was pelleted at 500×g (2 × 5 min), and 5% of the resultant supernatant was kept as the input fraction. Nuclei were pelleted at 4000×g (5 min), and the resulting supernatant centrifuged again at 17,000×g and kept as the cytoplasmic fraction. Nuclear pellets were washed twice in complete hypotonic buffer and dissolved in complete hypertonic buffer (15 mM HEPES, 400 mM KCl, 5 mM MgCl₂, 0.1 mM EDTA, 0.1% Tween 20, 10% glycerol, 1 mM DTT, complete EDTA-free proteinase inhibitor as above).

**Size-exclusion chromatography**. *C. elegans* lysate was prepared as described above for LC-MS/MS, and loaded onto a Superose 6 Increase 10/300 GL column. The column was equilibrated with Lysis buffer A and 1 ml fractions were collected using Unicorn 7.0 (GE Healthcare Life Sciences).

**Immunoblotting**. After SDS-PAGE using Bolt 4–12% Bis-Tris Plus gels (ThermoFisher Scientific), protein was transferred to PVDF membrane (0.45-micron pore size, ThermoFisher Scientific) using the XCell II Blot Module (ThermoFisher

Scientific). Membranes were blocked with 5% milk for 1 h, then incubated with primary antibody overnight at 4 °C, followed by secondary antibody for 1 h at RT. Unbound antibody was washed away with PBST (3 ×5 min), and SuperSignal West Pico PLUS Chemiluminescent Substrate (ThermoFisher Scientific) used for detection. The following commercially available antibodies were used: anti-FLAG M2-Peroxidase (A8592, Sigma A8592-2MG, diluted 1:1000), anti-Myc (9B11, Cell Signaling #2276, diluted 1:1000), anti-HA (C29F4, Cell Signaling #3724, diluted 1:1000), anti-V5 (A190, Bethyl Laboratories A190-120A, diluted 1:1000), anti-Histone H3 (Cell Signaling #9715, diluted 1:1000), anti-alpha tubulin (DM1A, abcam ab40742, diluted 1:4000), anti-Rabbit IgG (Bio-Rad #1706515, diluted 1:3000), and anti-Mouse IgG (Bio-Rad #1706516, diluted 1:3000). Western blots were imaged with Image Lab 4.1 (Bio-Rad).

**Yeast two-hybrid assays**. Yeast two-hybrid assays were performed by Panbionet (http://panbionet.com). cDNAs encoding NFKI-1 (full length) and NFKI-1 (1-374) were amplified by PCR and cloned into pGBKT7 vector, and cDNAs encoding MALT-1 (1-81) and MALT-1 (248-639) were amplified by PCR and cloned into pGADT7 vector (Clontech). Plasmids were transformed into the AH109 yeast strain, which expresses *HIS3* and *ADE2* reporters. Transformants were dropped separately onto SD-LW, SD-LWA and SD-LWH media containing 10 mM of 3-AT (3-amino-1,2,3-triazole), a competitive inhibitor of the HIS3 protein (His3p).

**RNA preparation**. 10 Gravid adults were allowed to lay eggs for 2 h on an OP50 lawn seeded 24 h previously, before being picked away. 8–10 plates were used per replicate, and all genetic backgrounds were prepared in parallel. Once animals reached late L4 stage, they were washed twice in M9 and frozen in liquid N₂. One milliliter Qiazol Lysis Reagent (Qiagen) and 300–400 µl 0.7 mm Zirconia beads (BioSpec) were added to worm pellets for mechanical disruption. Samples were disrupted with a TissueLyser (Qiagen), using 1 min at maximum power followed by 1 min on ice (repeated 4 times). RNA was extracted using the RNeasy Plus Universal Mini Kit (Qiagen), following the manufacturer's instructions with the exception that 1-Bromo-3-chloropropane (Sigma, B673) was used instead of chloroform.

Libraries were prepared using the TruSeq Stranded mRNA kit (Illumina) with polyA capture for mRNA, and sequenced on a HiSeq 4000 platform (Illumina) with 50 bp single-end reads. We sequenced five independent biological replicates for *npr-1 ilc-17.1*, and six for *npr-1, malt-1; npr-1,* and *npr-1 nfki-1*. Reads were aligned to the *C. elegans* genome using TopHat v2.1.0, and expression quantified using Cufflinks v2.2.1. For statistical comparisons a q-value <0.05 was considered significant. Only genes for which the FPKM (Fragments Per Kilobase of transcript per Million mapped reads) was ≥1 in *npr-1*, and for which the log2(fold change) between conditions was ≥0.25 were included in Fig. 8. Genes whose expression oscillates during development[85] were excluded.

GO terms for these genes were retrieved and GO enrichment calculated using g: Profiler (version e94_eg41_p11_592d917[86]). Terms with multiple testing corrected p-values <0.05 were considered enriched.

**Associative learning assay**. Chemotaxis assays were performed as previously described[87] with minor modifications. To establish salt gradients, 100 mM NaCl agar plugs were left overnight on assay plates containing 1 mM CaCl₂, 1 mM MgSO₄, 25 mM K₂HPO₄ pH 6. Plugs were removed immediately before the assay, and replaced with 1 µl 1 M NaN₃. Another 1 µl NaN₃ was added equidistant from the starting point as a control. The chemotaxis index was calculated as (A–B)/N, where A was the number of animals within 1 cm of the peak of the salt gradient, B was the number of animals within 1 cm of the control spot, and N was the number of all animals. Conditioning was performed as described previously[46]. Synchronized young adults raised on OP50 were washed three times in CTX buffer (5 mM K₂HPO₄ pH 6, 1 mM CaCl₂ and 1 mM MgSO₄), then left for 4 h on NGM agar with no NaCl (mock) or 300 mM NaCl (conditioned). For each transgenic strain the behavior of animals bearing the transgene was compared to that of their non-transgenic siblings.

**P. aeruginosa killing assays**. Slow killing assays were performed with 10 µM 5-fluorodeoxyuridine (FUdR)[88]. Synchronized L4 animals raised on OP50 were added to 0.35% peptone NGM plates, seeded the day before with PA14. Animals were scored every 12 h, and counted as dead if they did not respond to prodding. Logrank tests with Bonferroni correction were performed using OASIS (On-line Application for Survival Analysis, https://sbi.postech.ac.kr/oasis/[89]).

**Lifespan analyses**. Lifespan assays were performed on OP50, starting on day 1 of adulthood[90]. Scoring and statistical analyses were performed as described above for *P. aeruginosa* killing assays.

**Statistics**. Statistical tests and n values used for experiments in this paper are indicated in the corresponding figure legend or methods section. For salt chemotaxis and aggregation behavior, statistical significance between groups was determined using one-way ANOVA with Tukey's post hoc HSD. Differences in locomotory response and FRET levels (yellow cameleon) were evaluated by Mann-Whitney *U* test. Logrank tests with Bonferroni correction were used to compare

genotypes in lifespan and PA14 survival assays using OASIS (Online Application for Survival Analysis, https://sbi.postech.ac.kr/oasis/[89]).

**Reporting summary**. Further information on research design is available in the Nature Research Reporting Summary linked to this article.

## Data availability

Data supporting the findings of this study are provided as a Source Data file, and are available from the corresponding author. Full scans of blots are provided in Supplementary Fig. 10. RNA-seq data has been deposited in Gene Expression Omnibus (GEO) with accession number GSE144057. The mass spectrometry proteomics data have been deposited to the ProteomeXchange Consortium via the PRIDE partner repository with the dataset identifier PXD018000.

## Code availability

Zentracker, the custom MATLAB software used to analyze locomotory responses is available at https://github.com/wormtracker/zentracker. Neuron Analyzer, the custom-written MATLAB software used to analyze FRET levels (yellow cameleon), is available at https://github.com/neuronanalyser/neuronanalyser.

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

## Acknowledgements

We thank the *Caenorhabditis* Genetics Center (funded by National Institutes of Health Infrastructure Program P40 OD010440) and the Japanese knockout consortium for strains, the Cambridge Research Institute Genomics Core for whole genome sequencing, Adeline Colussi and Harvey McMahon for reagents, and de Bono lab members for comments on the manuscript. This work was supported by the Medical Research Council UK, the European Research Council (Advanced Grant 269058 to M.d.B), and Wellcome (209504/Z/17/Z Investigator Award to M.d.B.).

## Author contributions

S.F., C.C., M.A., S.B., and M.d.B. conceived experiments; S.F., C.C., M.A., and S.B. performed experiments; A.C. and G.N. performed sequence analysis; S.-Y.P.-C., F.B., and M.S. performed mass spectrometry analysis; S.F. and M.d.B. wrote the manuscript.

## Competing interests

The authors declare no competing interests.
