## [Peer Review File · Nature Communications]

Reviewers' comments:

Reviewer #1 (Remarks to the Author):

The study by Flynn et al investigates the conserved interleukin-17 (IL-17) signaling pathway using *Caenorhabditis elegans* as model organism. IL-17 is a pro-inflammatory cytokine which modulates neural circuit activity. IL-17 receptors are mainly expressed in the nervous system of *C. elegans* and IL-17 plays a role in sensing and avoidance behavior of high oxygen concentrations (21% O₂). The authors nicely combined biochemistry, genetics, and neuron-specific gene expression analyses to study the impact of IL-17 signals in neurons in the context of organismal physiology and behavioral state. They particularly focussed on the biochemical identification of signaling components that function downstream of IL-17 receptors. Therefore, the authors expressed epitope tagged proteins of the IL-17 pathway and performed immunoprecipitation experiments with respective worm lysates. Among other interesting binding partners, mass spectrometric analysis of the interacting proteins identified the paracaspase MALT-1. The authors carefully investigated the role of MALT-1 in context of additional interactors related to the IL-17 pathway. They nicely confirmed the biochemical data with further in vitro studies, genetic mutant analyses, and neuron-specific rescue experiments. Together, these results convincingly show a so far undescribed IL-17/MALT-1 axis important for escape behavior, associative learning, immunity, and longevity, which is likely conserved throughout evolution and will be influential for future studies in the field. Therefore, the manuscript is highly suitable for the readership of Nature comm.

Points to be addressed:

- 1.) The authors tested direct interaction between the recombinantly expressed proteins MALT-1 and NFKI-1, which nicely confirmed the in vivo data using worm lysates with tagged proteins. Since the interaction seems to be very robust in vitro it would be possible to use the same strategy to map the interaction site important for MALT-1/NFKI-1 binding. While this experiment is not necessary to further improve the quality of the work it would allow to perform additional rescue experiments with mutants lacking the binding site to investigate the signaling mechanism and its physiological impact.
- 2.) It is stated in the text that „MALT-1 provides a bridge between IL-17R associated complexes and NFKI-1“. The authors either tone down this conclusion or need to perform additional in vitro evaluation of complex formation by mixing all purified proteins together and characterizing binding sites.
- 3.) Similarly, the results of the gel filtration experiments is overinterpreted. The worm lysates used for these experiments contain many additional proteins which might bind to the tagged IL-17 signaling components (and additional interactors came out of the proteomics approaches). Therefore, any estimation about in vivo complex composition and/or size needs to be carefully discussed.

Reviewer #2 (Remarks to the Author):

Summary:

Flynn et al. use tandem biochemical and genetic methods to identify a conserved factor, the paracaspase MALT-1, as a mediator of interleukin receptor (ILR) signaling in the *C. elegans* nervous system. The manuscript contains a core of data that strongly support this conclusion. MALT-1 is identified as an interacting partner of ILR signaling factors. malt-1 mutants were

recovered from a screen that previously identified ILR and interleukin-like factors as modifiers of an oxygen-sensing circuit. Finally, genetic analysis places MALT-1 downstream of some ILR signaling partners and upstream of others. The discovery of MALT-1 as a mediator of ILR signaling is significant and, as the authors note, is likely conserved between *C. elegans* and vertebrates.

The manuscript contains additional data that suggest roles for MALT-1 in other signaling pathways and other behaviors. These data make the manuscript dense and, furthermore, introduce into the narrative many loose ends. A more compact manuscript that clarifies the few questions raised by the authors' analysis of MALT-1 in the oxygen-sensing circuit would be more accessible and would better highlight the overarching significance of the discovery of MALT-1 in ILR signaling.

Major concerns:

> The authors' data suggest that MALT-1 has multiple functions in different cells. The authors do not clearly enumerate these differences nor do they clearly discuss how a biochemical function of MALT-1 might explain all these different roles. This is a significant weakness of the manuscripts. Specifically, the authors show that MALT-1 is required in interneurons for proper calcium signaling in response to sensory stimuli but is also required in sensory neurons, which apparently do not require MALT-1 for calcium signaling. These data suggest at two distinct functions of MALT-1 in the nervous system. A third role is suggested by the observation that conserved residues likely required for enzymatic activity are required for MALT-1 to function in O₂-response behavior but are not required for MALT-1 to function in pathogen-tolerance. There is no clear synthesis of these data into a model that proposes a function for MALT-1. The model in Figure 7 is descriptive and only illustrates the authors' conclusion that MALT-1 is part of at least two distinct protein complexes.

> Data showing MALT-1 expression in the nervous system should be improved. Figure 2 is not very informative. It is difficult to see co-expression of MALT-1::mCherry and a marker of RMG fate in Figure 3. Likewise, it is difficult to see the expression of MALT-1::mCherry in some sensory neurons in Figure S5.

> The authors have generated an allele of *malt-1* that expresses a MALT-1::HA fusion. They should examine the expression of this fusion protein using immunohistochemistry to confirm that the *malt-1* reporter transgenes faithfully capture expression of endogenous MALT-1.

> It is not clear from the data in the manuscript whether loss of MALT-1 causes more general defects in behavior. Are locomotion, reproductive behaviors, and feeding behaviors grossly normal in these mutants?

> The authors should interpret with caution the appearance of ACTL-1 and PIK-1 in high molecular weight fractions from a gel filtration column (Figure 4i). This might represent aggregated protein or protein associated with unsolubilized membranes.

> The authors' conclude that MALT-1 is in a common pathway with ILCR-1 based on analysis of double mutants. They conclude that MALT-1 functions downstream of ACTL-1 and PIK-1 based on the ability of over-expression of MALT-1 to restore behavior to *pik-1* and *actl-1* mutants. Does over-expression of MALT-1 also bypass a requirement for ILCR-1 and ILC-17.1? The authors' model suggests that it will, but this is not tested.

> Survival in the presence of pathogenic *Pseudomonas* PA14 can be affected by behavior; this was previously shown by studies of *npr-1* variants. To rule out effects on survival via altered foraging/feeding behavior investigators have performed lifespan studies of animals grown on bacterial lawns that cover the entire plate. This eliminates the possibility of animals leaving the lawn and reducing their exposure to pathogen. The authors should test whether increased survival of *malt-1* mutants requires the possibility of leaving the lawn, for example using the 'big lawn'

protocol of Reddy, Kim and colleagues.

> The effect of malt-1 mutation on associative learning is not consistent with the authors' models and instead suggests a role for ILCR-1 that is independent of MALT-1. Specifically, loss of either ILCR-1 or ILCR-2 seems to cause a stronger defect than does loss of MALT-1. Do these data suggest yet another function for MALT-1? Is the MALT-1/TIR-1 pathway required for associative learning? As presented, these data do not clearly extend the major ideas put forth in the manuscript nor are they well integrated with other observations.

> The final figure presents a model of the complexes in which MALT-1 functions, which is useful. The figure also contains a summary of the effects of malt-1 mutation on behavior and physiology. This part of the figure has sketches of data that have unlabeled axes (confusing) and that summarize experimental results without synthesis.

> The discussion focuses on the authors' discovery that MALT-1 functions in a complex in ILCR-1 signaling. It might also be appropriate to discuss the role of the *C. elegans* nervous system in sensing pathogens and mediating immunity. This is a topic that interests many, but the discussion does not address it in any depth.

Other comments:

- > Authors should use standard gene/protein nomenclature throughout the manuscript.
- > p31 line 1 has typo: 'experiements'

Reviewer #3 (Remarks to the Author):

In this manuscript, Flynn and collaborators use a biochemical strategy to identify additional components of the IL-17 signalling pathway, which they previously characterized in the nematode *C. elegans* using genetic screens. They identified the *Ce* ortholog of MALT1 and provide convincing evidence that it is an important component required for IL-17 output. The characterization of MALT1 in *C. elegans* is novel and provides extremely interesting information on the evolution of this intensively-studied pathway. The *C. elegans* genome contains an ortholog of I kappa B but no obvious ortholog of NF kappa B. Although I am not an immunology specialist, I have the feeling that these results are important and raise interesting hypotheses to be tested in mammalian neurons. The manuscript is well written and provides high standard experimental data. Few points need, however, to be strengthened or clarified:

- interaction of NFKI-1 with the PIK-1-ACTL-1 complex:

as pointed out, no interaction between NFKI-1 and ACTL-1 or PIK-1 can be detected by LC-MS/MS or WB when using endogenously-tagged proteins (Fig. 4). The authors propose that this complex is only detected when increasing expression levels. One possibility would be that this complex is not detected because the proteins are in different compartments i.e. nuclear vs cytoplasmic when expressed at physiological levels. The fractionation experiment (Fig 5.a) is not convincing: there is a substantial contamination of the cytoplasmic fraction by histones. In addition, how many times was this experiment repeated? It would be much more convincing to label the different proteins by immunofluorescence and look for colocalization or possibly for in situ interaction using, for example, proximity ligation assay.

- lifespan:

raw data (number of replicates, number of animals excluded) and appropriate statistical tests (Kaplan Meir analysis and LogRank test) are needed in order to appreciate the reproducibility and the significance of the weak impact of the tested mutations on worm lifespan.

What is the lifespan of double mutants malt-1; ilc-17.1 ? Does the mutation of malt-1 can suppress the shorten lifespan of ilc-17.1 overexpressing worms? Those data would substantiate the potential role of MALT-1 as an effector of IL-17 for lifespan regulation

-MALT-1/TIR-1/MAPK-1 p15:

the authors showed that MALT-1 co-IPs with TIR-1 and thus investigated the functional significance of this association. The data obtained need to be further clarified and the relationship between MALT-1 and the IL17 receptors remains to be demonstrated in this context:

how tir-1 mutants behave as compared to malt-1 mutants in the same experimental conditions (Fig 6 and Sup 7)?

does MALT-1 really control MAPK-1 activity? Figure Sup7b shows a Western-Blot where only phospho-PMK-1 has been probed. The measurement of the ratio P-PMK-1/total PMK-1 is needed to conclude on the regulation of PMK-1 activity rather than the downregulation of pmk-1 expression. from fig.7a, MALT-1 appears to mostly act in the intestine. The contribution of neurons, and thus potentially of the IL7 receptor, remains to be further assessed. What is the immune response phenotype of double mutants malt-1; ilr-1 and ilcr-1; tir-1 ?

the authors reported that the expression of the anti-microbial peptide T24B8.5 and the phosphorylation of MPK-1, known to be respectively up and downregulated in tir-1 mutants, varied with the same trends in malt-1 mutants. However, while malt-1 mutants are resistant to PA infection, tir-1 inactivation completely suppressed this phenotype. What is the link between those effectors and the functional interaction between TIR-1 and MALT-1, if any?

minor point:

figure 2 is quite rudimentary and not very informative for a non-specialist. Higher quality picture and legends are required. What is the fluorescence visible in the pharynx? Is there any expression in non-neuronal tissues?

is the abbreviation "IP'd" commonly accepted? It is not extremely elegant.

Reviewer #1 (Remarks to the Author):

The study by Flynn et al investigates the conserved interleukin-17 (IL-17) signaling pathway using *Caenorhabditis elegans* as model organism. IL-17 is a pro-inflammatory cytokine which modulates neural circuit activity. IL-17 receptors are mainly expressed in the nervous system of *C. elegans* and IL-17 plays a role in sensing and avoidance behavior of high oxygen concentrations (21% O₂). The authors nicely combined biochemistry, genetics, and neuron-specific gene expression analyses to study the impact of IL-17 signals in neurons in the context of organismal physiology and behavioral state. They particularly focussed on the biochemical identification of signaling components that function downstream of IL-17 receptors. Therefore, the authors expressed epitope tagged proteins of the IL-17 pathway and performed immunoprecipitation experiments with respective worm lysates. Among other interesting binding partners, mass spectrometric analysis of the interacting proteins identified the paracaspase MALT-1. The authors carefully investigated the role of MALT-1 in context of additional interactors related to the IL-17 pathway. They nicely confirmed the biochemical data with further *in vitro* studies, genetic mutant analyses, and neuron-specific rescue experiments. Together, these results convincingly show a so far undescribed IL-17/MALT-1 axis important for escape behavior, associative learning, immunity, and longevity, which is likely conserved throughout evolution and will be influential for future studies in the field. Therefore, the manuscript is highly suitable for the readership of *Nature comm.*

We are pleased and encouraged that our Reviewer thinks the manuscript is highly suitable for *Nature Communications*.

Points to be addressed:

1.) The authors tested direct interaction between the recombinantly expressed proteins MALT-1 and NFKI-1, which nicely confirmed the *in vivo* data using worm lysates with tagged proteins. Since the interaction seems to be very robust *in vitro* it would be possible to use the same strategy to map the interaction site important for MALT-1/NFKI-1 binding. While this experiment is not necessary to further improve the quality of the work it would allow to perform additional rescue experiments with mutants lacking the binding site to investigate the signaling mechanism and its physiological impact.

We thank our reviewer for this suggestion. To address it, we first expressed sub-domains of MALT-1 and NFKI-1 in *E. coli*. We found that MALT-1 sub-domains expressed poorly, which made it difficult to obtain enough material for a co-IP experiment. We therefore attempted a different approach, using the yeast two-hybrid assay. We found that the Death domain of MALT-1 (aa 1-81) binds the N-terminus of NFKI-1 (Fig. 4i, see below).

		Bait		
		NFKI-1 (full)	NFKI-1 (1-374)	Vector
Prey	MALT-1 (1-81)	1	2	3
	MALT-1 (248-639)	4	5	6
	Vector		7	

Figure 4 MALT-1 has scaffolding and enzymatic roles in IL-17 signaling.

i Interaction of the MALT-1 Death Domain (1-81) with the N-terminus of NFKI-1 (1-374) in a yeast two-hybrid assay using nutritional selection (*ADE2*). All pair-wise tests were performed twice with similar results. Columns show 10-fold serial dilutions of each strain.

Specifically disrupting the interaction between MALT-1 and NFKI-1 *in vivo*, as suggested by our reviewer, would be an elegant way to test the physiological relevance of this interaction. However, ensuring we disrupt this interaction specifically, and leave the two proteins otherwise functionally intact, is non-trivial. It would require us to narrow down the region of the MALT-1 Death domain that binds NFKI-1 further, and then to identify point mutations that disrupt the interaction. We suspect that doing all this is likely to take considerable time, and to be pragmatic it may be better to do these experiments in a follow up manuscript.

2.) It is stated in the text that „MALT-1 provides a bridge between IL-17R associated complexes and NFKI-1“. The authors either tone down this conclusion or need to perform additional *in vitro* evaluation of complex formation by mixing all purified proteins together and characterizing binding sites.

We thank our reviewer for pointing this out. We agree that our choice of words was too strong here. We have rephrased our conclusion, and now say:

pg12:

MALT-1-HA immunoprecipitated NFKI-1-V5, and conversely NFKI-1-V5 immunoprecipitated MALT-1, supporting a direct physical interaction (Fig. 4g). MALT-1 also interacted directly with ACTL-1 (Fig. 4h).

3.) Similarly, the results of the gel filtration experiments is overinterpreted. The worm lysates used for these experiments contain many additional proteins which might bind to the tagged IL-17 signaling components (and additional interactors came out of the proteomics approaches). Therefore, any estimation about *in vivo* complex composition and/or size needs to be carefully discussed.

We agree with our Reviewer. While our observation of high molecular weight complexes of ACTL-1/PIK-1/MALT-1 in *ex vivo* extracts is consistent with these proteins forming a signalosome structure related to those formed by other Death domain signaling pathways, this is only one speculative interpretation. We have altered our wording to say this explicitly:

pg13:

The high-molecular weight species we observed may be an artefact of unsolubilized membrane or protein aggregation, or may represent interactions with additional proteins. Alternatively, they may report oligomeric complexes of ACTL-1/PIK-1/MALT-1 in *Ce* neurons related to the Myddosome and the CBM signalosome^{1,2}, although this hypothesis requires further testing.

Reviewer #2 (Remarks to the Author):

Summary:

Flynn et al. use tandem biochemical and genetic methods to identify a conserved factor, the paracaspase MALT-1, as a mediator of interleukin receptor (ILR) signaling in the *C. elegans* nervous system. The manuscript contains a core of data that strongly support this conclusion. MALT-1 is identified as an interacting partner of ILR signaling factors. *malt-1* mutants were recovered from a screen that previously identified ILR and interleukin-like factors as modifiers of an oxygen-sensing circuit. Finally, genetic analysis places MALT-1 downstream of some ILR signaling partners and upstream of others. The discovery of MALT-1 as a mediator of ILR signaling is significant and, as the authors note, is likely conserved between *C. elegans* and vertebrates.

The manuscript contains additional data that suggest roles for MALT-1 in other signaling pathways and other behaviors. These data make the manuscript dense and, furthermore, introduce into the narrative many loose ends. A more compact manuscript that clarifies the few questions raised by the authors' analysis of MALT-1 in the oxygen-sensing circuit would be more accessible and would better highlight the overarching significance of the discovery of MALT-1 in ILR signaling.

We thank Reviewer 2 for their comments, and for saying that the discovery of MALT-1 as a mediator of ILR signaling is significant and that the manuscript contains a core of data that strongly support this conclusion.

We understand our referee's suggestion to simplify our manuscript by excluding data that highlight roles for MALT-1 and IL-17 in regulating multiple *C. elegans* behaviors as well as physiology. We have struggled with this issue. Our goal in including some of these data is to highlight that IL-17 and MALT-1 signaling form an important neuroendocrine axis, not limited to oxygen sensing. We think bringing this point across to our readers is important, and may provoke similar experiments in other animal models. Our reviewer is quite right, however, that we have not tied up ends raised by these observations. This would involve substantial work without much altering the main message we wish to convey. We have sought to strike a balance, and have simplified our manuscript as much as possible, while retaining some reference to different phenotypes.

Major concerns:

The authors' data suggest that MALT-1 has multiple functions in different cells. The authors do not clearly enumerate these differences nor do they clearly discuss how a biochemical function of MALT-1 might explain all these different roles. This is a significant weakness of the manuscripts. Specifically, the authors show that MALT-1 is required in interneurons for proper calcium signaling in response to sensory stimuli but is also required in sensory neurons, which apparently do not require MALT-1 for calcium signaling. These data suggest at two distinct functions of MALT-1 in the nervous system.

We have sought to clarify this point. Our findings suggest the MALT-1 – NFKI-1 signaling ultimately regulates gene expression. Given this, we do not expect that this pathway would impinge on different neurons in same way. As is the case for other signaling pathways, the effects of signaling will depend on neural type and context. We have re-written our Discussion (pg 19) to clarify this point.

From Chen et al.³ Expression of ILCR-1 and ILCR-2 in partially rescues the speed responses of the respective mutants to 21% O₂.

A third role is suggested by the observation that conserved residues likely required for enzymatic activity are required for MALT-1 to function in O₂-response behavior but are not required for MALT-1 to function in pathogen-tolerance. There is no clear synthesis of these data into a model that proposes a function for MALT-1. The model in Figure 7 is descriptive and only illustrates the authors' conclusion that MALT-1 is part of at least two distinct protein complexes.

We think we may not have communicated our results clearly here. All functions of MALT-1 that we have examined require the protease active site of MALT-1 to be intact, including pathogen tolerance (see Fig. 6f and Supplementary Fig. 8b for the data).

> Data showing MALT-1 expression in the nervous system should be improved. Figure 2 is not very informative. It is difficult to see co-expression of MALT-1::mCherry and a marker of RMG fate in Figure 3. Likewise, it is difficult to see the expression of MALT-1::mCherry in some sensory neurons in Figure S5.

We apologize for our original images, which were below par. We have now provided better images (Fig. 3a and Supplementary Fig. 5).

Figure 3 MALT-1 mediates IL-17 signaling in RMG interneurons.

a A MALT-1::mCherry translational fusion, expressed from its endogenous promoter (4kb), is expressed in RMG interneurons. RMG is recognized by its characteristic shape, location, and using a *flp-5p::gfp* reporter (c, Kim and Li, 2004). Scale bars: 20µm.

Supplementary Figure 5. Related to Figure 2. MALT-1 is expressed in URX O₂-sensing neurons.

MALT-1::mCherry translational fusion, expressed from its endogenous promoter (4kb), is expressed in URX neurons in the head which are labelled with a *gcy-37p::gfp* reporter. Scale bars: 20µm.

> The authors have generated an allele of *malt-1* that expresses a MALT-1::HA fusion. They should examine the expression of this fusion protein using immunohistochemistry to confirm that the *malt-1* reporter transgenes faithfully capture expression of endogenous MALT-1.

This is a good experiment, and we have tried it multiple times. We could not however detect signal, at any stage. This is not an uncommon experience in the *C. elegans* community - immunohistochemistry signals from single copy reporters or knockins are often too weak to be detected. Almost the entire *C. elegans* community uses extrachromosomal, multicopy GFP transgenes to characterize gene expression. The good news is that these usually recapitulate endogenous expression patterns, as judged by phenotypic rescue experiments, cell- or tissue- specific RNA Seq experiments, and by analysing the data for the subset of genes where knocking in a fluorescent reporter gives a detectable signal.

> It is not clear from the data in the manuscript whether loss of MALT-1 causes more general defects in behavior. Are locomotion, reproductive behaviors, and feeding behaviors grossly normal in these mutants?

malt-1 mutants, like *ilcr-1*, *ilcr-2*, *actl-1*, *pik-1* and *nfki-1* mutants, or multiple mutant combinations thereof, have no obvious fertility defects as judged by population growth; they also mate well, grow at normal rates, move well, and show grossly normal feeding behavior. We quantified motility using a swimming assay, and observed a weak defect in *malt-1* mutants (Supplementary Figure 4e). Thus, *malt-1* mutants exhibit reduced locomotion, but this defect is subtle compared to the *malt-1* phenotype measured in assays of escape from 21% O₂. We now include this information in the text (pg 8).

Supplementary Figure 4. Related to Figure 1. MALT-1 promotes escape from 21 O₂.

e malt-1 mutants exhibit minor defects in thrashing rate ($n \geq 47$ animals). **, $P < 0.01$, ***, $P < 0.001$, ANOVA with Tukey's post hoc HSD.

> The authors should interpret with caution the appearance of ACTL-1 and PIK-1 in high molecular weight fractions from a gel filtration column (Figure 4i). This might represent aggregated protein or protein associated with unsolubilized membranes.

We agree that these results need to be interpreted with caution, and, prompted by our reviewer, we have altered our text accordingly. Additionally, as a control we checked that GAPDH does not appear in high-molecular weight fractions (Supplementary Figure 7).

pg13:

The high-molecular weight species we observe may be an artefact of unsolubilized membrane or protein aggregation, or may represent interactions with additional proteins. Alternatively, they may report oligomeric complexes of ACTL-1/PIK-1/MALT-1 in *Ce* neurons related to the Myddosome and the CBM signalosome^{1,2}, although this hypothesis requires further testing.

Supplementary Figure 7. Related to Figure 4. The elution profile MALT-1 and GAPDH proteins in *C. elegans* extract run on a Superose 6 Gel Filtration column and visualized by immunoblot. Unlike GAPDH control, MALT-1 is found in high-molecular weight fractions.

> The authors' conclude that MALT-1 is in a common pathway with ILCR-1 based on analysis of double mutants. They conclude that MALT-1 functions downstream of ACTL-1 and PIK-1 based on the ability of over-expression of MALT-1 to restore behavior to *pik-1* and *actl-1* mutants. Does over-expression of MALT-1 also bypass a requirement for ILCR-1 and ILC-17.1? The authors' model suggests that it will, but this is not tested.

We have now tested this. MALT-1 bypasses a requirement for ILCR-1 and ILC-17.1 (Fig. 5b).

Figure 5 MALT-1 and NFκI-1 provide partially parallel outputs of IL-17 signaling.

b Overexpressing *malt-1* gDNA restores the arousal response to 21% O₂ to *ilc-17.1* and *ilcr-1* mutants (n ≥ 52 animals). ***, P < 0.001, Mann-Whitney U test.

> Survival in the presence of pathogenic *Pseudomonas* PA14 can be affected by behavior; this was previously shown by studies of *npr-1* variants. To rule out effects on survival via altered foraging/feeding behavior investigators have performed lifespan studies of animals grown on bacterial lawns that cover the entire plate. This eliminates the possibility of animals leaving the lawn and reducing their exposure to pathogen. The authors should test whether increased survival of *malt-1* mutants requires the possibility of leaving the lawn, for example using the 'big lawn' protocol of Reddy, Kim and colleagues.

This is a very good point. We have now explicitly measured survival on PA14 *Pseudomonas aeruginosa* lawns that cover the entire plate, as suggested by our reviewer. The results show that the extended survival of *malt-1*, *ilcr-1* and *nfki-1* mutants is retained when animals cannot avoid the lawn (Fig. 6e-i).

Figure 6 MALT-1 acts downstream of IL-17 signaling to reprogram behavior and physiology.

e-h Mutants defective in *malt-1* and other IL-17 signaling components are resistant to *P. aeruginosa* PA14. The enhanced survival of *malt-1* mutants in PA14 “big lawn” assays compared to N2 controls is rescued by pan-neuronal expression of *malt-1* gDNA. ***, $P < 0.001$, logrank test ($n \geq 81$ animals).

i The enhanced resistance of *malt-1* mutants to PA14 requires TIR-1. Like *tir-1* mutants, *malt-1; tir-1* double mutants are hypersensitive to PA14 infection. ***, $P < 0.001$, logrank test ($n \geq 83$ animals).

> The effect of *malt-1* mutation on associative learning is not consistent with the authors' models and instead suggests a role for ILCR-1 that is independent of MALT-1. Specifically, loss of either ILCR-1 or ILCR-2 seems to cause a stronger defect than does loss of MALT-1. Do these data suggest yet another function for MALT-1? Is the MALT-1/TIR-1 pathway required for associative learning? As presented, these data do not clearly extend the major ideas put forth in the manuscript nor are they well integrated with other observations.

As we discuss briefly above, we appreciate that while our phenotyping data implicate IL-17 signaling in multiple circuits and paradigm, they do leave open ends. We have restructured the manuscript and amended the text and to reflect this (pg 15).

Previous work has shown that TIR-1 promotes forgetting in a salt associative learning assay related to, but not identical with, the assay we use. Prompted by our reviewer, we tested *tir-1* mutants and *malt-1* mutants in parallel. We could not detect a TIR-1 phenotype in our associative learning paradigm⁴, although in fairness our focus is on learning rather than forgetting. This means we do not know whether MALT-1 regulates the TIR-1 pathway in this context. We have downplayed the TIR-1 link.

> The final figure presents a model of the complexes in which MALT-1 functions, which is useful. The figure also contains a summary of the effects of *malt-1* mutation on behavior and physiology. This part of the figure has sketches of data that have unlabeled axes (confusing) and that summarize experimental results without synthesis.

We thank our reviewer for pointing this out. We have updated our model. The dual role of MALT-1 signaling in URX and RMG neurons is now illustrated (Fig. 7b) and the physiological effects of MALT-1 signaling are summarized (Fig 7c).

Figure 7 Model.

a Activation of nematode IL-17Rs ILCR-1 and ILCR-2 engages ACTL-1, the *Ce* ACT1-like adaptor, probably via their SEFIR domains. ACTL-1 recruits the *Ce* IRAK and MALT1 homologs to form the ACT1-IRAK-MALT1 signalosome in the cytoplasm. This serves a scaffolding function to recruit IκBζ/NFKI-1, and modulate its activity by an unknown mechanism. IκBζ, probably orchestrates changes in the transcriptome of RMG and other cells. MALT1-mediated cleavage of unknown substrate(s) positively regulates IκBζ signaling. In parallel to this pathway, MALT-1 forms a complex of unknown function with TIR-1/SARM1, and with multiple RNA-binding proteins.

b ILCR receptors and downstream signaling components including MALT-1 are expressed in many neurons. This neuronal signaling cassette alters associative learning as well as O₂-escape behaviors, and suppresses lifespan and immunity.

> The discussion focuses on the authors' discovery that MALT-1 functions in a complex in ILCR-1 signaling. It might also be appropriate to discuss the role of the *C. elegans* nervous system in sensing pathogens and mediating immunity. This is a topic that interests many, but the discussion does not address it in any depth.

We agree that we were terse in discussing the importance of the *C. elegans* nervous system in mediating immunity. This is an area of obvious interest, and we have amplified our discussion to take account of this (pg 19).

Other comments:

> Authors should use standard gene/protein nomenclature throughout the manuscript.

We have gone through the manuscript to ensure we use standard nomenclature.

> p31 line 1 has typo: 'experiements'

We thank our reviewer for pointing out this inadvertent mistake.

Reviewer #3 (Remarks to the Author):

In this manuscript, Flynn and collaborators use a biochemical strategy to identify additional components of the IL-17 signalling pathway, which they previously characterized in the nematode *C. elegans* using genetic screens. They identified the *Ce* ortholog of MALT1 and provide convincing evidence that it is an important component required for IL-17 output. The characterization of MALT1 in *C. elegans* is novel and provides extremely interesting information on the evolution of this intensively-studied pathway. The *C. elegans* genome contains an ortholog of I kappa B but no obvious ortholog of NF kappa B. Although I am not an immunology specialist, I have the feeling that these results are important and raise interesting hypotheses to be tested in mammalian neurons. The manuscript is well written and provides high standard experimental data. Few points need, however, to be strengthened or clarified:

We are pleased that our reviewer finds our manuscript to be well-written and to have high standard experimental data.

- interaction of NFKI-1 with the PIK-1-ACTL-1 complex:
as pointed out, no interaction between NFKI-1 and ACTL-1 or PIK-1 can be detected

by LC-MS/MS or WB when using endogenously-tagged proteins (Fig. 4). The authors propose that this complex is only detected when increasing expression levels.

We thought it possible that endogenous MALT-1-NFKI-1 complexes were not detected in our *ex vivo* lysates due to a high dissociation rate coupled with dilution of the proteins in the lysate relative to *in vivo* concentrations (which would make the complex unlikely to reform). To reduce the time available for complex dissociation we shortened the IP step of our protocol (from 3 - 4h to 30 minutes). Using these conditions to IP endogenously tagged proteins, we detected an interaction between NFKI-1 and MALT-1: NFKI-1 was enriched in MALT-1 IPs compared to control IgG IPs (Fig. 4d). We performed three biological replicates, each with similar results. This supports the hypothesis that an endogenous MALT-1/NFKI-1 complex exists *in vivo*.

Figure 4 MALT-1 has scaffolding and enzymatic roles in IL-17 signaling.

d Endogenous ACTL-1, PIK-1 and NFKI-1 co-IP with endogenous MALT-1 in *npr-1* animals. Anti-HA antibody was used to immunoprecipitate MALT-1 complexes. Half of the lysate was immunoprecipitated with anti-IgG as a control. Tags were knocked in by CRISPR.

One possibility would be that this complex is not detected because the proteins are in different compartments i.e. nuclear vs cytoplasmic when expressed at physiological levels. The fractionation experiment (Fig 5.a) is not convincing: there is a substantial contamination of the cytoplasmic fraction by histones. In addition, how many times was this experiment repeated?

We have repeated the fractionation experiment five times, and have now indicated this in our revised manuscript. The reviewer is correct that some replicates contain some histone contamination in the cytoplasmic fraction. However, we do not think this impacts the inference we draw from the experiments, namely that much of NFKI-1 is nuclear. Sub-cellular fractionation is rarely absolutely clean, but we have reduced the contamination in our nuclear fractions (Fig. 5a).

Figure 5 MALT-1 and NFKI-1 provide partially parallel outputs of IL-17 signaling.

a Immunoblot analysis of IL-17 signaling components from nuclear and cytoplasmic fractions of *Ce* lysate. I, input, C, cytosolic, N, nuclear. NFKI-1 is predominately nuclear; ACTL-1 and MALT-1 are distributed between the nucleus and cytoplasm.

It would be much more convincing to label the different proteins by immunofluorescence and look for colocalization or possibly for in situ interaction using, for example, proximity ligation assay.

This is a very sensible suggestion. We spent significant time trying to get immunofluorescence experiments to work, both before our initial submission, and during revision. However, we cannot detect fluorescence signal from the epitope-tagged single copy knockins. This is not an uncommon observation in the *C. elegans* field – GFP or epitope knockins only reveal detectable (immune)fluorescence signals for genes that are highly expressed. Almost everyone in the field does their experiments using multi-copy transgenes (and very few people look for biochemical interactions in *ex vivo* extracts). Our reviewer's suggestion of using proximity labeling is a very good one, but this method has not been optimized for *C. elegans*, so setting it up would be a substantial effort.

We agree with our reviewer that differences in the sub-cellular localization of MALT-1 and NFKI-1 may limit the interaction between these proteins. Nevertheless, the IP data we have now added suggests that there is a physical interaction between the endogenous proteins.

- lifespan:

row data (number of replicates, number of animals excluded) and appropriate statistical tests (Kaplan Meir analysis and LogRank test) are needed in order to appreciate the reproducibility and the significance of the weak impact of the tested

mutations on worm lifespan.

We have included these data in our revised manuscript. Supplementary Table 6 provides a summary of (i) the number of animals counted and excluded (ii) the number of biological replicates and (iii) the p-value for each relevant comparison. Additionally, Kaplan-Meier analysis, and logrank tests are shown in Supplementary Table 8.

What is the lifespan of double mutants *malt-1*; *ilc-17.1* ? Does the mutation of *malt-1* can suppress the shorten lifespan of *ilc-17.1* overexpressing worms? Those data would substantiate the potential role of MALT-1 as an effector of IL-17 for lifespan regulation

The lifespan of *malt-1*; *ilc-17.1* double mutants is not significantly different from either single mutant, suggesting these genes function in the same pathway to regulate longevity (Fig. 6l). Consistent with this, disrupting *malt-1* suppresses the shortened lifespan phenotype of animals overexpressing ILC-17.1 (Fig. 6m).

Figure 6 MALT-1 acts downstream of IL-17 signaling to reprogram behavior and physiology.

l The phenotypes of *malt-1* and *ilc-17.1* mutants are not additive ($n \geq 111$ animals).

m The shortened lifespan of animals overexpressing ILC-17.1 is abolished in *malt-1* mutants ($n \geq 92$ animals).

-MALT-1/TIR-1/MAPK-1 p15:

the authors showed that MALT-1 co-IPs with TIR-1 and thus investigated the functional significance of this association. The data obtained need to be further clarified and the relationship between MALT-1 and the IL17 receptors remains to be demonstrated in this context:

how *tir-1* mutants behave as compared to *malt-1* mutants in the same experimental conditions (Fig 6 and Sup 7)? ainly

As our reviewer points out, although the physical interaction between MALT-1 and TIR-1 is striking and intriguing, the functional significance of this interaction is unclear, and requires further study. We have now stated this explicitly in our revised manuscript.

does MALT-1 really control MAPK-1 activity? Figure Sup7b shows a Western-Blot where only phospho-PMK-1 has been probed. The measurement of the ratio P-PMK-1/total PMK-1 is needed to conclude on the regulation of PMK-1 activity rather than the downregulation of *pmk-1* expression.

We found an antibody that allowed us to measure total PMK-1 protein levels, and find these were similar in *malt-1* and WT animals. However, to simplify the manuscript, and because we see some variability in our results, we removed these data from our manuscript.

from fig.7a, MALT-1 appears to mostly act in the intestine. The contribution of neurons, and thus potentially of the IL7 receptor, remains to be further assessed.

We presume our reviewer is referring here to Supplementary Figure 7A (Supplementary Fig. 9a in the revised manuscript). Our original labeling of this figure was not very clear. In this figure we show that expressing MALT-1 either in the intestine, using the *ges-1* promoter, or pan-neuronally, using the *rab-3* promoter, substantially rescues the T24B8.5p::GFP expression defect. In the revised manuscript we have explicitly labeled the figure with 'intestinal expression' and 'pan-neuronal expression'.

What is the immune response phenotype of double mutants *malt-1; ilcr-1* and *ilcr-1; tir-1* ?

To address this point, we quantitated expression of the *T24B8.5p::gfp* transgene in *malt-1; ilcr-1*, *malt-1; tir-1*, and *ilcr-1; tir-1* double mutants. Our results are consistent with several models, so we have not sought to draw strong conclusions from them, other than to say that the expression of the T24B8.5 reporter is reduced in both *malt-1* and *ilcr-1* mutants.

Supplementary Figure 9. Related to Figure 6. MALT-1 promotes p38 MAPK signaling.

a Expression from *T24B8.5p::GFP*, a reporter of the innate immune response, is inhibited in *malt-1(db1194)*, *ilcr-1(tm5866)*, and *tir-1(tm3036)* mutants compared to N2. $n \geq 35$ animals. *, $P < 0.05$, **, $P < 0.01$, ***, $P < 0.001$, ANOVA with Tukey's post hoc HSD.

The authors reported that the expression of the anti-microbial peptide T24B8.5 and the phosphorylation of MPK-1, known to be respectively up and downregulated in *tir-1* mutants, varied with the same trends in *malt-1* mutants. However, while *malt-1* mutants are resistant to PA infection, *tir-1* inactivation completely suppressed this phenotype. What is the link between those effectors and the functional interaction between TIR-1 and MALT-1, if any?

As we mention earlier, our motivation in presenting some of these phenotypic data was to highlight that IL-17 signaling influences multiple *C. elegans* phenotypes, which we think is interesting. However, we have not unravelled the network of interactions underlying each phenotype. We have downplayed the TIR-1 link, and now explicitly say that the functional importance of the biochemical interaction between TIR-1 and MALT-1 remains to be investigated.

minor point:

figure 2 is quite rudimentary and not very informative for a non-specialist. Higher quality picture and legends are required. What is the fluorescence visible in the pharynx? Is there any expression in non-neuronal tissues?

We have provided a better image, showing expression of MALT-1::GFP in the nervous system and pharynx.

Figure 2 MALT-1 is expressed widely in the nervous system.

A transgene expressing C-terminally GFP-tagged MALT-1 from its endogenous promoter (4kb of upstream DNA) fluorescently labels much of the nervous system, including many neurons in the head (red box) and tail (blue

box). MALT-1::GFP expression is also seen throughout the pharynx. White arrows point to neurons, arrowheads point to the pharyngeal bulbs.

Is the abbreviation "IP'd" commonly accepted? It is not extremely elegant.

We agree that the shorthand form 'IP'd' is ugly, with little to recommend it except brevity. We have replaced 'IP'd' with 'immunoprecipitated' throughout the manuscript.

1. Lin, S.-C., Lo, Y.-C. & Wu, H. Helical assembly in the MyD88-IRAK4-IRAK2 complex in TLR/IL-1R signalling. *Nature* **465**, 885–890 (2010).
2. Qiao, Q. *et al.* Structural Architecture of the CARMA1/Bcl10/MALT1 Signalosome: Nucleation-Induced Filamentous Assembly. *Molecular Cell* **51**, 766–779 (2013).
3. Chen, C. *et al.* IL-17 is a neuromodulator of *Caenorhabditis elegans* sensory responses. *Nature* **542**, 43–48 (2017).
4. Saeki, S., Yamamoto, M. & Iino, Y. Plasticity of chemotaxis revealed by paired presentation of a chemoattractant and starvation in the nematode *Caenorhabditis elegans*. *Journal of Experimental Biology* **204**, 1757–1764 (2001).

Reviewers' comments:

Reviewer #1 (Remarks to the Author):

The authors fully addressed my remaining suggestions. The manuscript convincingly describes the importance of the IL-17/MALT-1 axis especially for escape behavior, associative learning, and longevity, which is of key interest for the readership of Nature comm.

Thorsten Hoppe

Reviewer #2 (Remarks to the Author):

Flynn et al. have made significant changes to this manuscript and addressed many of my concerns. The manuscript would benefit from some further editing, and I do have some remaining questions.

Major concerns

> No statistical analysis of IP-MS data is shown in Fig. 1. It is not clear that SEM is an informative statistic given the nature of the data and the low number of replicates. The authors should present these data in context of all the data acquired from the experiment. For example, the authors could show volcano plots and indicate proteins of interest as they did in Fig. 4.

> In panels 4e and 4f the authors should make clear what alleles of malt-1 and pik-1 are used for IP/MS analysis of NFKI-1 complexes. The authors should also clarify whether they detected any peptides derived from malt-1 or pik-1 in these experiments.

> It would be very helpful if the position and nature of malt-1 mutations were added to the malt-1 gene model in Fig. S2.

> When describing the data in Fig. 5a the authors describe MALT-1 as being present in nuclear and in cytosolic fractions while NFKI-1 as predominantly in nuclear fractions. It is not clear that this conclusion can be made. There is little NFKI-1 immunoreactivity in the fractions and it is possible that if more material were loaded NFKI-1 signal would be detected in cytosolic fractions.

> In the discussion the authors speculate that their studies reveal the function of the ancestral MALT-1 complex (P20 L6-9). The basis for this is not clear.

> The discussion wanders and brings up points that are left as loose ends. It should be more focused on a smaller number of points that relate to the data. For example the authors mention that an antimicrobial peptide is downregulated in malt-1 mutants despite their being resistant to pathogen. The authors also mention the possibility that MALT-1 can regulate gene expression post-transcriptionally. And the authors refer to nuclear hormone receptors that so-precipitate with MALT-1 but are not the subject of the studies being discussed.

Minor issues

> P4 L12: remove the note-to-self and add the reference

> P6 L4: remove the phrase 'non-physiological' ; this is not warranted

> P7 and on: 'Ce' is used as shorthand for C. elegans. This is not standard.

> P11 L11: 'MALT-1 signals as a protease' is not warranted - replace w/ 'MALT-1 protease function is required...'

> P12 L1: 'NFK-1 was detectable only after short IPs....' - where are the data supporting this statement?

> P19 L10: Please change the sentence 'These differential effects of IL-17 signaling are not too surprising.'

> P19 L11: typo: 'regulate genes expression'

Reviewer #3 (Remarks to the Author):

The authors satisfactorily addressed all my concerns.

Reviewer #1 (Remarks to the Author):

The authors fully addressed my remaining suggestions. The manuscript convincingly describes the importance of the IL-17/MALT-1 axis especially for escape behavior, associative learning, and longevity, which is of key interest for the readership of Nature comm.

Thorsten Hoppe

We are very grateful to Dr Hoppe for taking the time to review our manuscript.

Reviewer #2 (Remarks to the Author):

Flynn et al. have made significant changes to this manuscript and addressed many of my concerns. The manuscript would benefit from some further editing, and I do have some remaining questions.

We have done some further editing and sought to address remaining questions.

Major concerns

> No statistical analysis of IP-MS data is shown in Fig. 1. It is not clear that SEM is an informative statistic given the nature of the data and the low number of replicates. The authors should present these data in context of all the data acquired from the experiment. For example, the authors could show volcano plots and indicate proteins of interest as they did in Fig. 4.

As suggested by our reviewer we now show complete IP-MS datasets in Fig. 1. Fig. 1f-i show experiments that are representative of multiple biological replicates.

Figure 1 MALT-1 forms a complex with ACTL-1, PIK-1/IRAK and NFKI-1.

b-i Pull-down of ACTL-1-FLAG, PIK-1-Myc or NFKI-1::GFP specifically co-IPs MALT-1 (**b-g**). Conversely, pull-down of MALT-1::GFP specifically co-IPs ACTL-1, PIK-1 and NFKI-1 (**h and i**). Total spectral counts, a semi-quantitative readout of abundance⁷⁷, are shown. (**c, e, g, and i**) as in **b, d, f, and h** except showing only the region marked by the black box in **b, d, f, and h** respectively. (**f-i**) Data is representative of two (**f and g**), or three (**h and i**) biological replicates.

> In panels 4e and 4f the authors should make clear what alleles of *malt-1* and *pik-1* are used for IP/MS analysis of NFKI-1 complexes. The authors should also clarify whether they detected any peptides derived from *malt-1* or *pik-1* in these experiments.

Fig. 4e and 4f were performed using the *malt-1(db1194)* and *pik-1(tm2167)* alleles. We have added this information to the figure legend.

Peptides derived from MALT-1 and PIK-1 were detected in these experiments. We have now listed these in Supplementary Table 2. As depicted in the volcano plots the relative abundance of the peptides corresponding to MALT-1 and PIK-1 was significantly reduced in the corresponding mutants compared to WT.

> It would be very helpful if the position and nature of *malt-1* mutations were added to the *malt-1* gene model in Fig. S2.

We have added this information to Fig. S2.

Supplementary Figure 2. Related to Figure 1. MALT1.

a MALT1 paracaspase domain organization. Black arrows indicate the impact of *malt-1* mutations. DD, death domain; Ig, Immunoglobulin-like fold.

> When describing the data in Fig. 5a the authors describe MALT-1 as being present in nuclear and in cytosolic fractions while NFKI-1 as predominantly in nuclear fractions. It is not clear that this conclusion can be made. There is little NFKI-1 immunoreactivity in the fractions and it is possible that if more material were loaded NFKI-1 signal would be detected in cytosolic fractions.

We agree with our reviewer that it is possible that NFKI-1 may be present in our cytoplasmic fractions at levels that are below our detection limit. We have indicated this in the text.

pg13:

ACTL-1-FLAG and MALT-1-HA were consistently detected in both cytoplasmic and nuclear fractions (Fig. 5a). NFKI-1-V5 was predominantly in nuclear fractions (Fig. 5a; five replicates), although as NFKI-1-V5 immunoreactivity in the fractions was weak we cannot rule out the possibility that NFKI-1 was also present in the cytoplasmic fractions at levels below our detection threshold.

> In the discussion the authors speculate that their studies reveal the function of the

ancestral MALT-1 complex (P20 L6-9). The basis for this is not clear.

As suggested by our reviewer, we have clarified the basis for this speculation.

pg 20:

MALT1-like paracaspases are found in organisms lacking other CBM components¹⁸, suggesting MALT1 has unknown functions that predate its coaction with Bcl10 and CARD domain proteins. Our results raise the possibility that one ancestral function was in IL-17 signaling. As IL-17Rs are found throughout metazoa⁶⁵, we speculate that the ACTL-1-IRAK-MALT-1 complex we have identified is the original and primary mechanism by which IL-17Rs signal in non-amniote animals, from cnidarians to cephalochordates. In amniotes, ACT1 orthologs have lost a death domain (DD) that is present in ACT1 orthologs from most other lineages⁶⁵. DDs mediate homotypic interactions in large immune complexes such as the Myddosome⁶⁶, and are present in both MALT1 and IRAKs. The DD – SEFIR domain architecture of ACT1 present in non-amniotes resembles the DD – TIR domain structure of MyD88, since TIR and SEFIR domains are related⁶⁷.

> The discussion wanders and brings up points that are left as loose ends. It should be more focused on a smaller number of points that relate to the data. For example the authors mention that an antimicrobial peptide is downregulated in malt-1 mutants despite their being resistant to pathogen. The authors also mention the possibility that MALT-1 can regulate gene expression post-transcriptionally. And the authors refer to nuclear hormone receptors that so-precipitate with MALT-1 but are not the subject of the studies being discussed.

We thank the reviewer for this suggestion. We have removed reference to interaction partners of MALT-1 that were not the subject of our study, focusing our discussion on the primary conclusions.

Minor issues

> P4 L12: remove the note-to-self and add the reference

We thank the reviewer for pointing out this omission, which we have corrected.

> P6 L4: remove the phrase 'non-physiological' ; this is not warranted

We have removed this 'non-physiological', which we agree is redundant.

> P7 and on: 'Ce' is used as shorthand for *C. elegans*. This is not standard.

We have replaced all instances of *Ce* with *C. elegans*.

> P11 L11: 'MALT-1 signals as a protease' is not warranted - replace w/ 'MALT-1 protease function is required...'

We thank our reviewer for pointing this out, and have changed the wording with: 'MALT-1 protease activity is important for its function.....'

> P12 L1: 'NFK-1 was detectable only after short IPs....' - where are the data

supporting this statement?

We were able to detect a physical interaction between MALT-1 and NFκB-1 only after reducing the IP time from 3 - 4h to 30 minutes. However, as these experiments were performed on different days and are therefore not directly comparable we have removed this statement.

> P19 L10: Please change the sentence 'These differential effects of IL-17 signaling are not too surprising.'

We have changed this sentence to:

'These different effects of IL-17 signaling may be indicative of cell-type specific effects on gene expression'.

> P19 L11: typo: 'regulate genes expression'

We thank our reviewer for pointing out this mistake, which we have corrected.

Reviewer #3 (Remarks to the Author):

The authors satisfactorily addressed all my concerns.

We thank the reviewer for the time spent reviewing our manuscript.

REVIEWERS' COMMENTS:

Reviewer #2 (Remarks to the Author):

The authors' revisions address my comments.

Reviewer #2 (Remarks to the Author):

The authors' revisions address my comments.

We thank the reviewer for their time spent reviewing our manuscript.